# Multi-omic analysis of SDHB-deficient pheochromocytomas and paragangliomas identifies metastasis and treatment-related molecular profiles

Aidan Flynn[1], Andrew D. Pattison[1], Shiva Balachander[1], Emma Boehm[1], Blake Bowen[1], Trisha Dwight[2], Fernando J. Rossello[1,3,4,5], Oliver Hofmann[1], Luciano Martelotto[1], Maia Zethoven[6], Lawrence S. Kirschner[7], Tobias Else[8], Lauren Fishbein[9], Anthony J. Gill[10,11], Arthur S. Tischler[12], Thomas Giordano[8], Tamara Prodanov[13], Jane R. Noble[14], Roger R. Reddel[14], Alison H. Trainer[6,15], Hans Kumar Ghayee[16], Isabelle Bourdeau[17], Marianne Elston[18], Diana Ishak[19], Joanne Ngeow Yuen Yie[19,20], Rodney J. Hicks[21], Joakim Crona[22], Tobias Åkerström[23], Peter Stålberg[23], Patricia Dahia[24], Sean Grimmond[1], Roderick Clifton-Bligh[2,10,25] ✉, Karel Pacak[13,25] ✉ & Richard W. Tothill[1,15,25] ✉

Hereditary *SDHB*-mutant pheochromocytomas (PC) and paragangliomas (PG) are rare tumours with a high propensity to metastasize although their clinical behaviour is unpredictable. To characterize the genomic landscape of these tumours and identify metastasis biomarkers, we perform multi-omic analysis on 94 tumours from 79 patients using seven molecular methods. Sympathetic (chromaffin cell) and parasympathetic (non-chromaffin cell) PCPG have distinct molecular profiles reflecting their cell-of-origin and biochemical profile. *TERT* and *ATRX*-alterations are associated with metastatic PCPG and these tumours have an increased mutation load, and distinct transcriptional and telomeric features. Most PCPG have quiet genomes with some rare co-operative driver events, including *EPAS1*/HIF-2α mutations. Two mechanisms of acquired resistance to DNA alkylating chemotherapies are identifiable; *MGMT* overexpression and mismatch repair-deficiency causing hypermutation. Our comprehensive multi-omic analysis of *SDHB*-mutant PCPG therefore identifies features of metastatic disease and treatment response, expanding our understanding of these rare neuroendocrine tumours.

Pheochromocytomas (PC) and paragangliomas (PG) are heritable neuroendocrine neoplasms arising from the sympathetic and parasympathetic nervous system. They are clinically remarkable for catecholamine hypersecretion (e.g., dopamine, noradrenaline, adrenaline) causing morbidity and occasionally death from cardiovascular sequelae[1,2]. PCPG are typically slow growing, but metastases develop in 10–20% of patients[3]. Current biomarkers are inadequate for predicting progression to metastatic disease, which is currently incurable.

Metastatic risk, which guides surveillance recommendations[4–6], is highest in patients carrying constitutional pathogenic variants in

*SDHB*. SDHB-deficiency and cellular succinate accumulation inhibits prolyl hydroxylase domain proteins (PHD1/2) leading to oxygen-independent stabilization of hypoxia inducible factor alpha (HIF-α) and HIF target gene activation - a phenomenon known as pseudohypoxia[7,8]. Succinate excess also inhibits other 2-oxoglutarate-dependent dioxygenases, including TET2 and jumonji histone lysine demethylases, causing genome-wide hypermethylation and epigenetic reprogramming[8,9]. While the causative role of SDHB-deficiency in metastatic PCPG is well founded, other genomic changes also likely contribute, such as *ATRX* and *TERT* mutations[10–16]. However, most PCPG genomic studies to date have used whole-exome sequencing, limiting detection of non-coding variants, structural alterations, telomeric features and DNA mutation patterns.

Herein, we conduct an integrated multi-omic analysis of an internationally sourced and clinically well-annotated cohort of *SDHB*-mutant PCPG including parasympathetic head and neck PG (HN-PG) - a relatively understudied group to date. Importantly, paired primary and metastatic tumours are analysed in a subset of cases. We make important observations with respect to cell-of-origin, treatment response and clinical outcome and confirmed the presence of *TERT/ATRX* mutations and other features in metastatic cases.

## Results

### Whole genome and multi-omic analysis of *SDHB*-associated PCPG

To characterize the genomic landscape of *SDHB*-mutant PCPG, we performed multi-omic analysis of an international (A5 consortium) cohort representing 94 primary and/or metastatic PCPG tumours from 79 patients (Fig. 1A) (*see* Supplementary Data 1 and 2 for clinical details). Analysis included whole-genome sequencing (WGS) of tumour and matched blood ($n = 94$), whole transcriptome sequencing (WTS) ($n = 91$), small-RNA seq ($n = 90$), DNA methylation profiling ($n = 93$) and C-circle analysis ($n = 89$). Droplet-based (10x) single nuclei (sn)RNA-seq and snATAC-seq was applied in a subset of cases (Fig. 1B).

Review of histopathology and clinical data confirmed disease diagnosis in all cases. Paired synchronous/metachronous primaries were analysed in four cases (E128, E136, E159, E229) and clonal independence confirmed by discordant somatic profiles (Supplementary Fig. 1 and 2). With respect to clinical behaviour, 40 tumours from 37 patients were ostensibly non-metastatic with at least twelve-months of clinical follow-up (median: 60 months, range 12–456). An additional eight primary PCPG from eight patients had less than 12 months of clinical follow-up without metastases reported (termed: short clinical follow-up). Thirty-four of 79 patients had confirmed metastatic disease. Twenty-eight primary PCPG were analysed from 26 patients who developed metastases and in six patients at least one paired metastasis was available, confirming the clonal link between the primary and metastatic tumours (termed: metastatic primary) (Fig. 1C). For 21 primary PCPG from 20 patients, metastasis was reported but metastatic tissue was not available (termed: primary - metastasis reported) (Fig. 1C). Finally, in eight patients, one or more metastases were analysed but a primary tumour was not analysed.

WGS confirmed pathogenic germline *SDHB* mutations consisting of missense ($n = 41$), nonsense ($n = 21$), frameshift insertions/deletions ($n = 6$), large deletion events ($n = 7$), and donor splice-site mutations ($n = 4$) (Fig. 1D). Two recurrent large deletion events were identified in two or more unrelated patients (chr1:g.17052136-17054668_del, chr1:g.17048756-17064432_del). WGS showed variable length homologous DNA sequence at the breakpoint ends of a subset of the large deletions, implicating homologous recombination as a potential cause of the germline deletions (Supplementary Fig. 3). Somatic loss-of-heterozygosity (LOH) at 1p36.13 with or without somatic copy-loss was detected in all tumours and SDHB immunohistochemistry confirmed tumoural SDH-deficiency.

## Molecular profiling confirms genotype-subtype and cell-of-origin transcriptional profile

Molecular subtyping is known to reflect the genotypic features of PCPG[13,17–21]. To confirm genotype-phenotype relationships among A5 tumours we applied UMAP clustering to WTS ($n = 91$), small-RNA-seq ($n = 90$) and DNA methylation ($n = 93$) datasets. Published RNA-seq data across a spectrum of PCPG genotypes was included for comparison[13,14,20]. Here, we used the C1/C2 annotation based on seven PCPG gene-expression subtypes previously defined by single-nuclei and bulk gene-expression analysis[22]. As expected, all A5 *SDHB*-mutant tumours clustered among C1A (SDHx) tumours by WTS (Fig. 2A). UMAP of the small-RNA-seq data showed a similar clustering pattern; however, four A5 tumours from two patients clustered with an outlier group consisting of unrelated genotypes (Fig. 2B). This small-RNA-seq outlier group included C2Bi (*MAX*) PCPG that frequently exhibit loss-of-heterozygosity of chr14q and silencing of the imprinted *DLK1-MEG3* miRNA cluster located on chr14q31-32[20] (Supplementary Fig. 4). By DNA methylation profiling, *SDHB*-mutant PCPG were distinguished from other PCPG genotypes, with the exception of one tumour (E229-P1) (Fig. 2C), and *SDHB*-mutant tumours showed genome-wide hypermethylation (Supplementary Fig. 5), consistent with a prior report[9]. Unsupervised clustering therefore confirmed the expected molecular profile of the A5 *SDHB*-mutant tumours as well as the pathogenicity of constitutional *SDHB* variants.

UMAP clustering of only the A5 tumours showed clear separation of parasympathetic (non-chromaffin cell) HN-PG from sympathetic (chromaffin cell) PCPG (Fig. 2D–F). Two mediastinal dopamine-secreting PG (E148-P1, E155-P1) clustered with the HN-PG group, suggesting these tumours may be non-chromaffin cell paragangliomas[23] while one HN-PG (E185-P1) and one abdominal-thoracic PG (E128-P2) showed variable clustering. Supervised differential expression analysis between HN-PG and sympathetic PCPG identified 4615 genes and 36 microRNAs (adjusted *p*-value < 0.05, log fold change >1) (Supplementary Data 3 and 4). HN-PG and the suspected non-chromaffin mediastinal tumours had low expression of the chromaffin cell marker *CARTPT*[24], neural transcriptional regulators (*TFAP2B, TOX3, GATA3, POU4F, NEUROG2, PAX2*), *HOX* genes (*HOXA1-10, HOXB4, HOXB6-9, HOXC4-HOXC13*), and the long non-coding RNA *HOTAIR* (Fig. 2H, Supplementary Fig. 6). Similar to the *HOX* gene clusters, clustered genes on chr12q24.32 including *TMEM132C* and lncRNAs were also lowly expressed. Concordant DNA methylation patterns were also observed (Fig. 2G, I, Supplementary Fig. 7).

HN-PG are predominantly biochemically silent and accordingly catecholamine biosynthesis genes including *TH* and *DBH* were lowly expressed (Fig. 2J). Although most sympathetic PCPG are biochemically active, a small number of A5 cases had normal catecholamine levels and such tumours also had low *TH* expression. Some patients with HN-PG/non-chromaffin cell tumours and sympathetic PCPG had elevated plasma 3-methoxytyramine (3-MT) (a dopamine metabolite[25,26]) and these tumours had low *DBH* expression - DBH being necessary to convert dopamine to norepinephrine. Low *TH* and *DBH* expression corresponded with gene-promoter methylation (Supplementary Fig. 8). Meanwhile, variable expression of the norepinephrine transporter (*SLC6A2*) was observed with gene-promoter methylation corresponding with low expression in sympathetic PCPG (Supplementary Fig. 9). Low *SLC6A2* (NET) expression is clinically relevant since NET is required for uptake of theranostic agents e.g.,[123/124/131]I-meta-iodobenzylguanidine (MIBG)[27,28].

With respect to neurotransmitters, *TPH1* (tryptophan hydroxylase), essential for serotonin (5-HT) synthesis, was overexpressed in the HN-PG/non-chromaffin cell tumours (Fig. 2G, Supplementary Fig. 10). TPH is known to be expressed in glomus cells of the carotid body (presumed to be the HN-PG cell-of-origin)[29], and 5-HT storage vesicles have been observed in some HN-PG, although urinary 5-HIAA is not elevated in most HN-PG patients[30]. Meanwhile, while

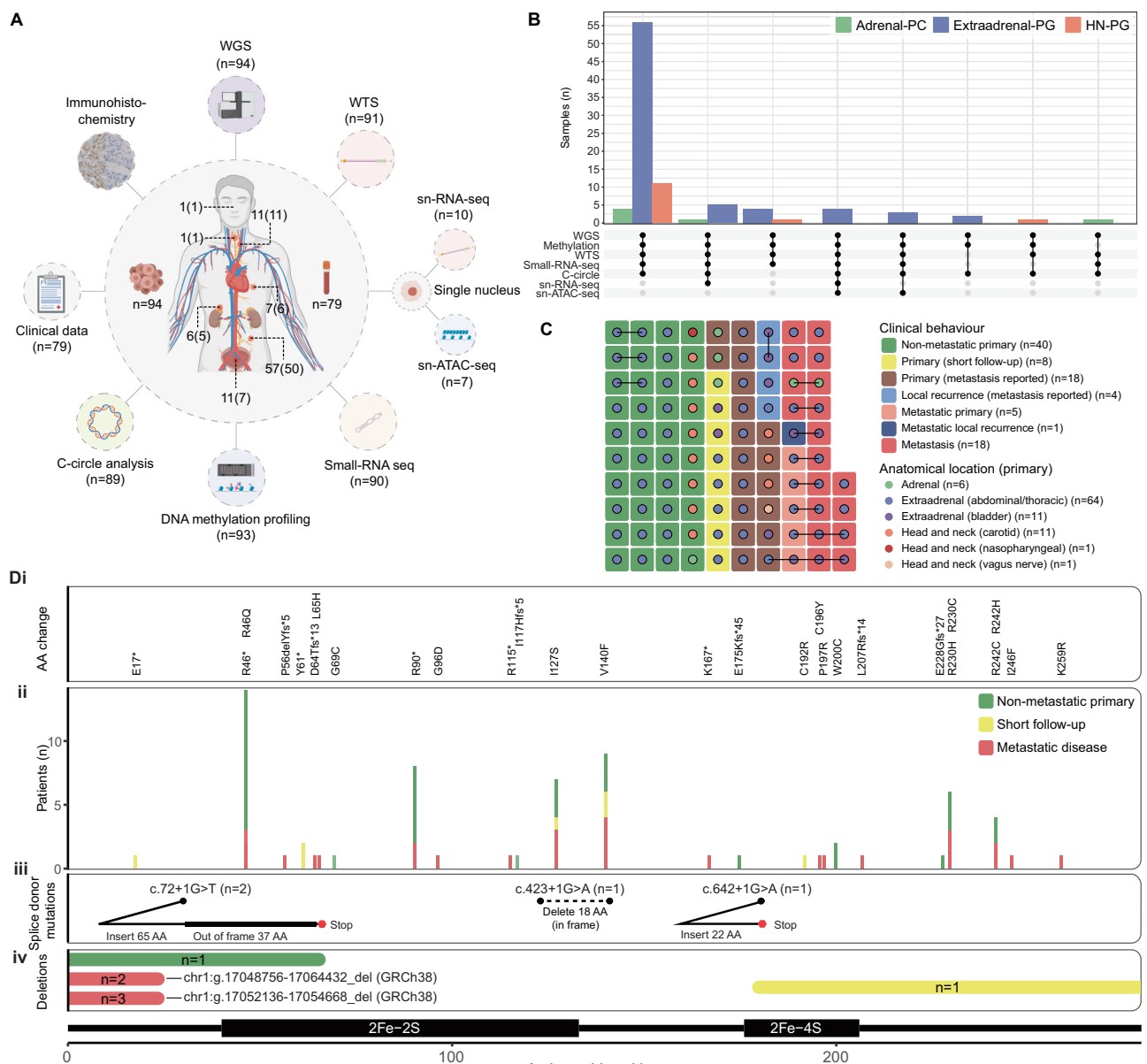

**Fig. 1 | Multi-omic analysis of SDHB-mutated PCPG. A** Outer ring: Overview of the analytical methods applied to the cohort annotated with the number of tumours analysed. Inner ring: The number of tumours and patients (in parenthesis) analysed from each anatomical location. Where a metastasis was analysed, the location of the primary tumour is indicated. Created with components from BioRender. (Licence: Tothill, R. (2025), https://BioRender.com/y24d952). **B** A summary of the combination of analytical methods applied to each tumour with respect to anatomical primary location. The upper panel indicates the total number of samples from each anatomical location analysed with the respective combination of assays (lower panel). **C** Anatomical location (dot colour) and clinical behaviour (tile colour) of each tumour analysed. Paired samples from the same patient are joined by a line. **D** Summary of the germline *SDHB* mutations detected by WGS across the cohort and their respective position within the protein amino acid sequence. (i) Amino acid changes for single nucleotide variants and small insertion/deletion events (NP_002991.2). (ii) The total number of patients observed with each amino acid change where bar colour indicates the clinical disease course. (iii) Schematic of the consequence of splice donor mutations. (iv) Regions of the SDHB protein sequence deleted by large scale structural events. Bar colour indicates the clinical disease course and the number of patients affected is indicated within the bar.

immunohistochemical detection of choline acetyltransferase (*CHAT*) has been reported in HN-PG[31], *CHAT* mRNA was not overexpressed in these tumours (Supplementary Fig. 11).

### Genome-wide mutational features of *SDHB*-deficient PCPG

Consistent with prior studies, therapy naive PCPG ($n = 84$) had a low mutation burden (median = 0.32, range 0.03–4.24 mut/Mb)[11,13,14,20]. Single-base substitution (SBS) signature analysis (COSMICv3) showed predominantly age-related or clock-like mutational signatures (SBS1, SBS5) (Fig. 3A). With respect to doublet base substitutions (DBS) the predominant patterns were DBS 2, 4, 9 and insertion/deletion (ID)

signatures ID 5, 8 and 9. With respect to systemic cytotoxic treatments, 10/94 (12%) of tumours were resected post-treatment (Supplementary Fig. 12) and chemotherapy-related mutational signatures were observed in some cases including SBS11 (DNA alkylation) (discussed below).

PCPG tumours showed recurrent patterns of chromosomal changes similar to observations in previous studies[13,14,18] including loss of chr1p, chr2q, chr3, chr8p, chr11, chr17p and chr22 (GISTIC q < 0.05) (Fig. 3B, Supplementary Fig. 13). Significant copy-number gains were observed on chr1q in sympathetic (chromaffin cell) PCPG, while parasympathetic (HN-PG/non-chromaffin cell) tumours had frequent loss

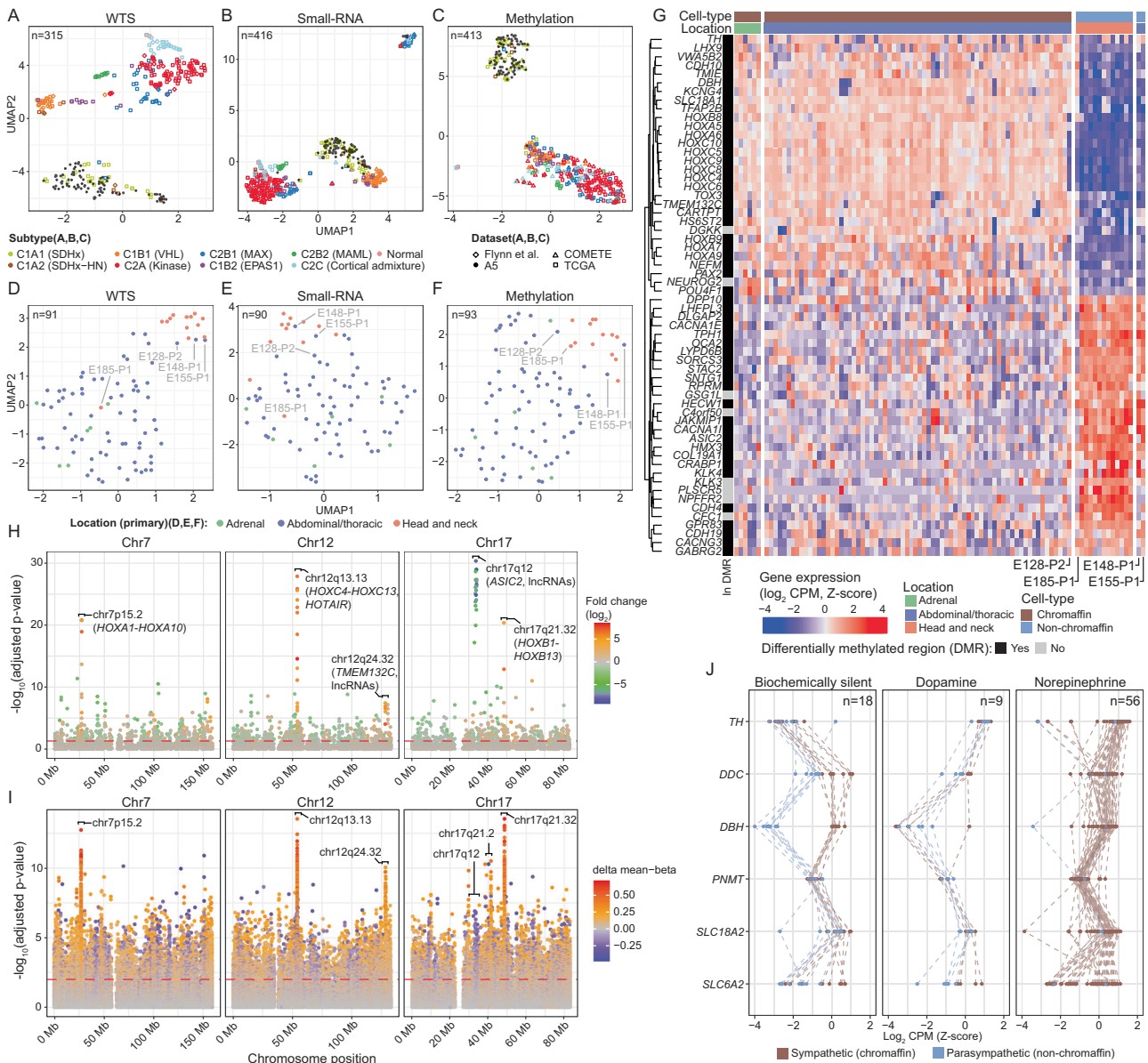

**Fig. 2 | Genomic profiling separates sympathetic and parasympathetic PGL.**
UMAP dimensional reduction was used to cluster WTS (**A**, n = 315), small-RNA-seq (**B**, n = 416) and DNA methylation data (**C**, n = 413) with previously published data[9,13,14,20,22,88]. UMAP clustering was also repeated for respective A5 data types in isolation (**D**–**F**, n = {**D**: 91, **E**: 90, **F**: 93}). **G** Differential expression profiling was performed between abdominal-thoracic PCPG and HN-PG. The heatmap shows CPM (log₂, z-score) expression values for each tumour (x-axis) for the top differentially expressed genes (limma moderated t-test Benjamini-Hochberg adjusted p-value < 0.05, ranked by log-fold-change, top and bottom 30 genes are shown)(y-axis). The top annotation bars indicate the suspected cell-of-origin based on UMAP clustering and the anatomical location of the tumour, respectively. The left annotation bar indicates whether the gene was also in a differentially methylated region

for the same contrast. **H, I** Spatial distribution of adjusted p-values (y-axis, -log10, RNA-seq p-values from limma Benjamini-Hochberg corrected moderated t-test, methylation p-values derived using the RUV-inverse method from the missMethyl package) along chromosomes 7, 12, and 17 (x-axis) from differential expression (**H**) and probe-level differential methylation analysis (**I**) between sympathetic PCPG (n = 75) and HN-PG (n = 12). **J** Expression of catecholamine biosynthesis and processing pathway genes. Line colour indicates the anatomical location of the tumour and sub-panels segregate tumours based on which catecholamines were above upper normal limit during clinical testing. Expression data for the A5 cohort were combined with a larger compendium of publicly available[22] data representing the different PCPG subtypes (n = 315) before values were normalized to Z-scores. Only A5 tumours are shown.

of chr1q. A minority of tumours had chromothripsis (n = 6) or genome-doubling (n = 9), as previously reported in PCPG[13,14,20]. Notably, we identified significant chr7 gains in the metastatic group (GISTIC q < 0.05). Sub-clonal copy number events were observed in several samples by WGS and copy number inference from sn-RNA-seq (Supplementary Fig. 14A–D), with primary tumours exhibiting greater clonal diversity than metastases (One-sided Student's t-test, p < 0.01) (Supplementary Fig. 14E).

Few genes were recurrently altered and only 15 cancer-related genes had >1 predicted pathogenic mutation using Cancer Genome

Interpreter[32] (Fig. 3C, Supplementary Data 5). Consistent with prior studies, *TERT* and *ATRX* were the most frequently mutated genes[11,13,15] (discussed in detail below). *ATRX* mutations were significant drivers using the dN/dS[33] method while *TERT* promoter (p*TERT*) mutations were significant using OncoDriveFML[34]. Two tumours had somatic *EPAS1* (HIF-2α) pathogenic variants near codon 531 (prolyl) targeted by PHD2 (Fig. 3D). One *EPAS1* mutation (E230-P1, NP_001421.2:p.Tyr532-Cys) was previously described in a PCPG patient with polycythemia[35,36], while a second *EPAS1* mutation (E183-P1, NP_001421.2:p.Asp539Gly) involved the same codon as a previously reported *EPAS1* pathogenic

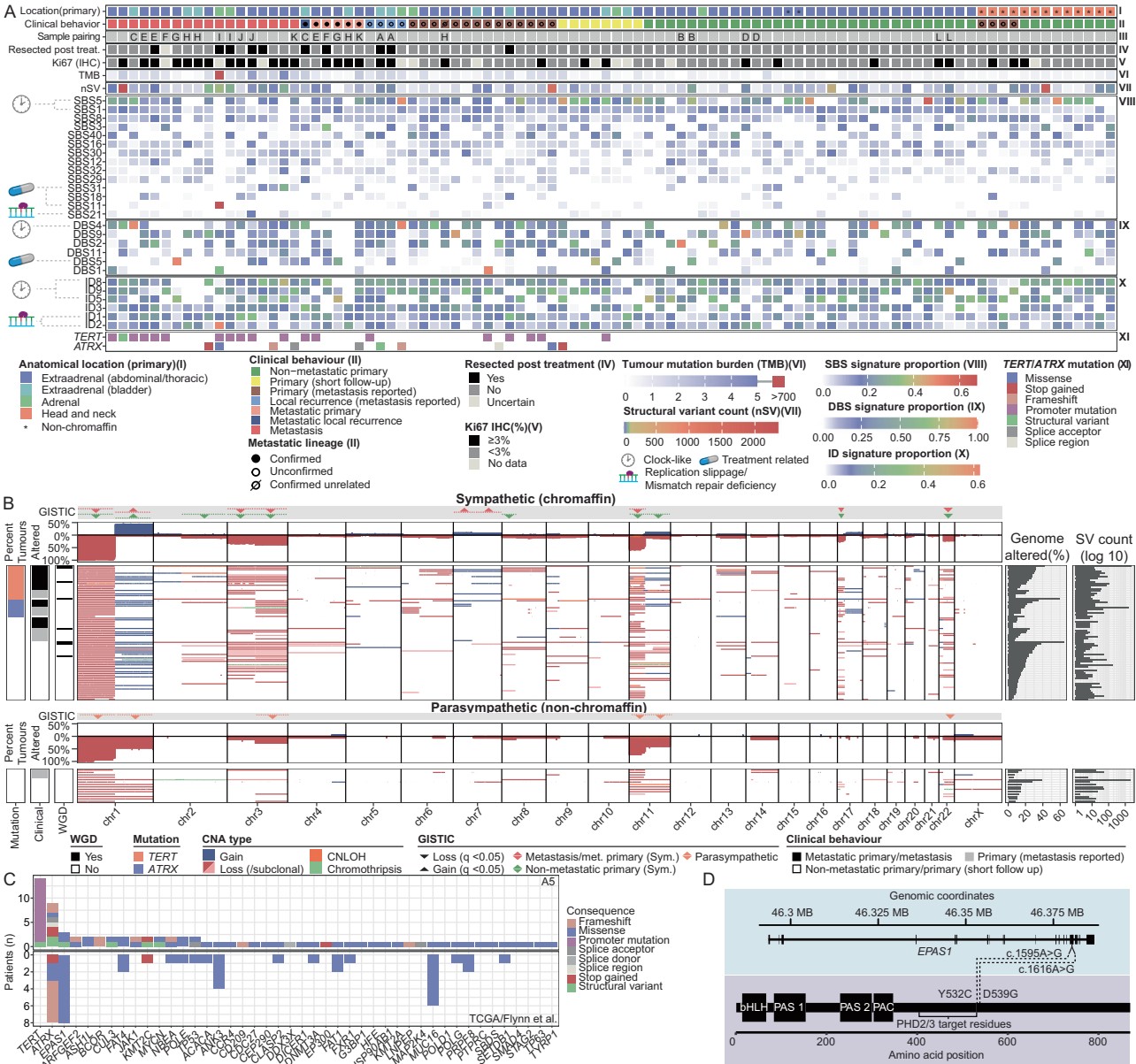

**Fig. 3 | Genomic and clinical features of PCPG. A** (i) Location of the primary tumour. An asterisk indicates whether the tumour had a non-chromaffin expression signature. (ii) Clinical behaviour. Primary tumours from patients with metastatic disease are annotated as to whether the tumour was able (filled circle) or unable (open circle = no metastasis sequenced, strike-through = confirmed unrelated) to be confirmed as the metastatic clone through sequencing of a paired metastasis. (iii) Identifier linking samples from the same patient. (iv) Indicates if the sample was resected after cytotoxic treatment. (v) Immunohistochemical scoring of Ki67; greater than 3% positive cells is indicated by a black square. (vi) Tumour mutation burden (mutations per megabase) and (vii) number of structural variants are indicated using a heatmap. (viii–x) Signature analysis was performed using the MutationalSignatures package. Signatures that contributed 15% of mutations and at least 500, 50, or 10 mutations for SBS, ID, and DBS signatures, respectively, in at least one tumour are shown. Signatures are ranked (top to bottom) based on the

mean proportional contribution. Colour indicates the proportion of mutations attributable to each signature (xi) *TERT/ATRX* mutation status. **B** Aggregate and per-sample copy number aberrations in sympathetic (rows 2 and 3, respectively) and parasympathetic (rows 5 and 6, respectively) PCPG. Statistically significant recurrently deleted (downward arrow) or amplified (upwards arrow) regions are indicated (rows 1 and 4). Clinical behaviour and whole genome doubling are annotated to the left of per-sample data (rows 3 and 5). The percentage of genome altered and structural variants count are annotated on the right. Losses and gains as well as percent genome altered are relative to sample ploidy rather than diploid. **C** Genes that were recurrently altered and predicted to be drivers by Cancer Genome Interpreter across the A5 cohort (top) and the respective number of mutations observed in published datasets (bottom). **D** Schematic showing somatic mutations observed in *EPAS1*.

variant[37]. Other recurrently mutated cancer genes included chromatin modifiers *KMT2C (n = 2)* and *BCOR (n = 2)*. *TP53* somatic mutations were found in three metastatic tumours consistent with a known low frequency in PCPG[13,14,20]. Finally, a *MYCN* hotspot mutation was found in one tumour (E129-P1, NP_005369.2:p.Pro44Leu), as previously reported in another study[38].

## Somatic *TERT* mutations, structural rearrangement and promoter methylation

Hotspot p*TERT* mutations (chr5:1295113 G > A, chr5:1295135 G > A, GRCh38) were identified in 14 metastatic cases (17.7%). Targeted deep sequencing of p*TERT* excluded potential sub-clonal mutations in the other cases. snATAC-seq data showed a promoter peak central to the

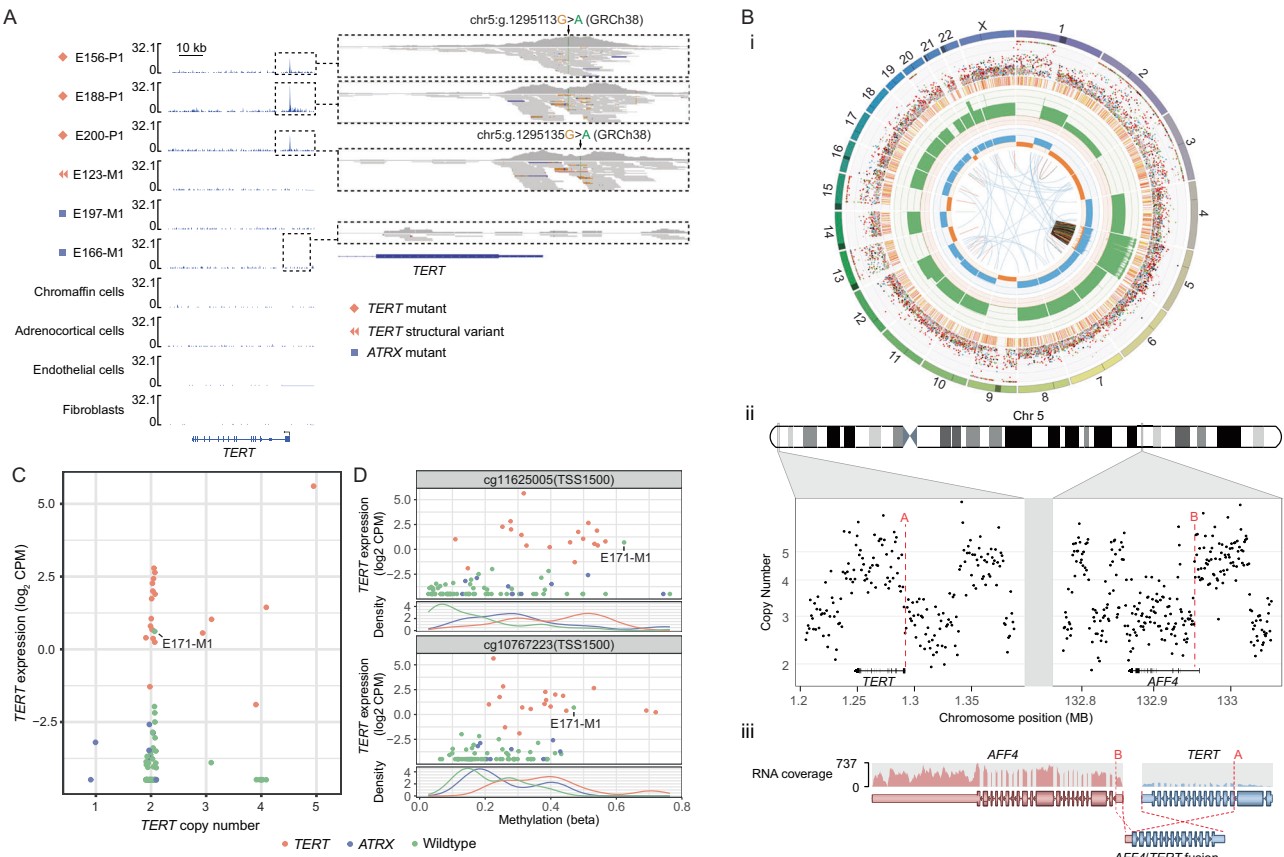

**Fig. 4 | TERT promoter mutations and structural alterations. A** sn-ATAC-seq covering the *TERT* gene and promoter (inset) region. Data is shown for tumour cells from two *ATRX* mutant tumours, three *TERT* mutant tumours, and one tumour with a *TERT* structural variant. Normal chromaffin, adrenocortical, fibroblast, and endothelial cells are shown for contrast. **B** (i) Circos plot describing somatic alterations detected by WGS in E123-M1. Outer ring indicates the chromosome, second outer ring marks SNVs (blue: C > A, black: C > G, red: C > T, grey: T > A, green: T > C, pink: T > G) and Indels (yellow: insertions, red: deletions). Third outer ring indicates total copy number, (green: >2, red: <2), the fourth outer ring indicates the copy number of the minor allele (blue: >1, orange: <1). The centre circle

displays structural variants. (ii) Copy number segmentation data from the *AFF4* and *TERT* gene regions in E123-M1 showing multiple short segmental copy-number changes due to chromothripsis. The break points contributing to an *AFF4-TERT* fusion are marked with red dashed lines. (iii) Schematic of *AFF4-TERT* fusion detected by WTS. **C** *TERT* expression (y-axis, log₂ CPM) versus gene copy number at the TERT locus (x-axis). **D** *TERT* expression (y-axis, log₂ CPM) versus the methylation status (x-axis) of probes in the *TERT* promoter and *TERT* hypermethylated oncological regions (rows 1 and 3). Density plot of probe methylation beta-values (rows 2 and 4). Point and line colour indicate *TERT/ATRX* mutation status.

de novo ETS binding site[39] in three p*TERT* mutants tested and all snATAC-seq reads had the somatic variant, confirming open chromatin (Fig. 4A). Chromothripsis of chromosome 5 created an *AFF4-TERT* fusion in one case (E123) (Fig. 4B). WGS supported a DNA breakpoint in *AFF4* 3-prime of untranslated exon 1 and DNA sequence 390 bp upstream of the *TERT* transcription start site (Fig. 4Bii). RNA-seq split-reads supported a fusion transcript involving *AFF4* exon 1 spliced to *TERT* exon 3 (Fig. 4Biii). This validates our prior report of *TERT* structural rearrangements in PCPG[16].

All *TERT* altered cases had elevated *TERT* expression with the highest expression observed in the *AFF4:TERT* fusion tumour. *TERT* copy-number gain did not correlate with *TERT* overexpression in the absence of *TERT* mutation or fusion events (Fig. 4C). We found *TERT* promoter hypermethylation including the THOR domain (probes cg11625005, cg10767223) in *TERT* altered PCPG, as previously described[15] (Fig. 4D). *TERT* promoter hypermethylation and concordant overexpression was also observed in a single metastatic *TERT* wild-type tumour (E171-M1) (Fig. 4D).

### *ATRX* alterations are associated with the ALT phenotype
Pathogenic *ATRX* mutations included nonsense and splice-site mutations, small frameshift insertions or deletions, and a large deletion in one case (Fig. 5A). One missense *ATRX* mutation (unknown significance) was

detected (NM_000489.6:c.6778 C > T, NP_000480.3:p.H2260Y). *ATRX* expression was low in most *ATRX* altered cases, presumably due to nonsense-mediated decay (Supplementary Fig. 15).

Alternative lengthening of telomeres (ALT) is a known feature of *ATRX* altered PCPG and other cancer types[12]. C-circle analysis was done to predict ALT activity[40] and C-circles were detected in all *ATRX* altered and one *TERT* altered primary tumour (E143-P1) (discussed below) (Fig. 5B). As expected, the telomeric tumour:blood ratio, a proxy for tumour telomere length, was higher in *ATRX* altered compared to *TERT* altered or *TERT/ATRX* wild-type tumours (Two-sided Student's t-test *p*-value < 0.00025) with no significant difference in telomeric DNA content observed between the latter two groups. Long telomeres in the absence of C-circles were detected in four *ATRX/TERT* wild-type PCPG, including two HN-PG.

ALT features can include aberrant telomeric variant repeat (TVR) usage, intrachromosomal insertion of telomeric DNA and expression of telomeric repeat-containing RNA (*TERRA*)[41]. Relative TVR usage was altered in C-circle positive versus negative tumours (Fig. 5C). When normalized to total read count (rather than telomeric read count) only three TVRs (TTAGGG, GTTGGG, GTAGGG) remained significant after false-discovery rate correction (Benjamini & Hochberg, *p* < 0.1). When normalized to the telomeric reads the canonical TVR (TTAGGG) was proportionally more abundant in C-circle positive compared to

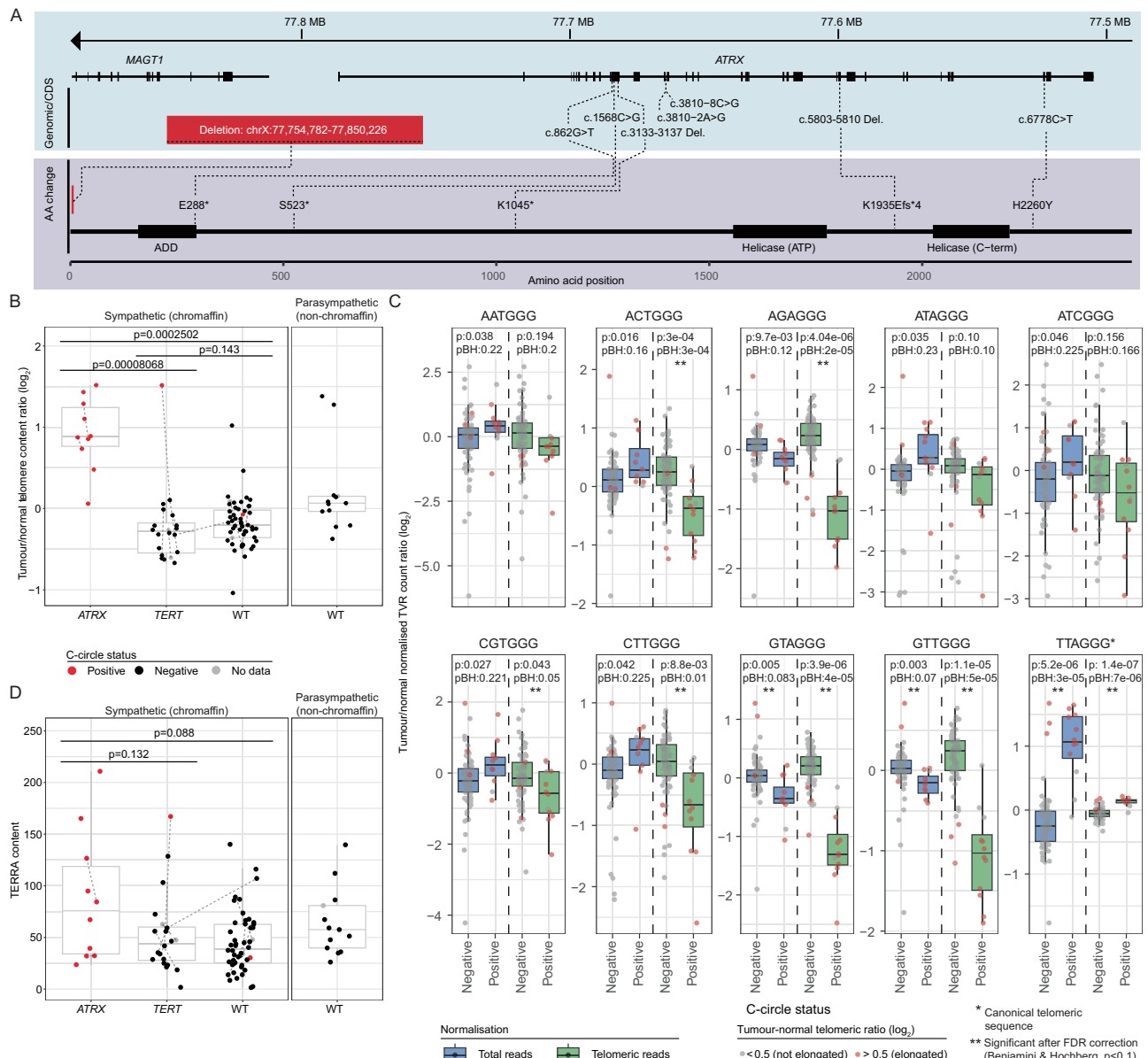

**Fig. 5 | Loss of *ATRX* leads to telomere dysregulation. A** Genomic (top) and protein (bottom) position of alterations in *ATRX*. **B** Tumour/normal ratio of telomere read content (y-axis) in relation to *ATRX*[mut] ($n = 10$), *TERT*[mut] ($n = 20$) and wild-type ($n = ${Chromaffin: 48, Non-chromaffin:14}) status (x-axis). Presence (red) or absence (black) of C-circles is indicated by dot colour. Tumours from the same patient are joined by a line. Values were averaged across paired tumours and a two-sided Student's t-test was used to test for differences. **C** Telomere variant repeats (TVRs) of the type NNNGGG were detected in WGS from tumour and matched normal using TelomereHunter. Count values were normalized against intratelomeric (green) or total (blue) read count. The tumour/normal ratio of normalized counts (y-axis) are shown with respect to the presence ($n = 12$) or absence ($n = 76$) of detected c-circles (x-axis). Only TVRs which were significantly different using a two-tailed Students t-test ($p < 0.05$) applied to values normalized to total reads are shown. TVRs that were significant after false-discovery correction (Benjamini & Hochberg, $p < 0.1$) are indicated with a double-asterisk. *P*-values before (p) and after (pBH) correction are shown. Data points are coloured to indicate a tumour/normal telomere content ratio (log$_2$) greater than (red) or less than (grey) 0.5. **D** Reads containing telomeric sequences were counted from WTS data using TelomereHunter. The telomeric content (y-axis) was computed as the number of unmapped reads containing telomeric sequence times 1,000,000 divided by the total number of reads with a GC content similar to telomeric repeats (n = {*TERT*[mut]: 20, *ATRX*[mut]: 10, Chromaffin[wildtype]: 45, Non-chromaffin[wildtype]: 14}). Data point colours, x-axis, and statistical testing are as described in (**B**). The hinges of each boxplot correspond to the first and third quartiles and the median value is marked. The whiskers extend to the largest and smallest value no greater than 1.5 times the interquartile range above or below the upper and lower hinges, respectively.

negative tumours. Only one intrachromosomal telomeric insertion was detected across the cohort, occurring in *ATRX* altered case E198-M1. *TERRA* expression was elevated in *ATRX* altered tumours; however, the difference did not reach statistical significance (Two-sided Student's t-test $p > 0.05$)(Fig. 5D). It should be noted that the tag counting method of telomere quantitation cannot differentiate reads originating from centromeric and telomeric regions.

## *TERT/ATRX* alterations are associated with metastatic progression and late somatic events

*TERT/ATRX* alterations were found in 21/34 (61%) metastatic cases and significantly associated with metastasis (Pearson's Chi-squared *p*-value 9.349e-08). Only one non-metastatic tumour (E156-P1) had a p*TERT* mutation; however, this case had a short clinical follow-up (Fig. 6A). Evidence for late acquisition of p*TERT* mutations was found in two cases:

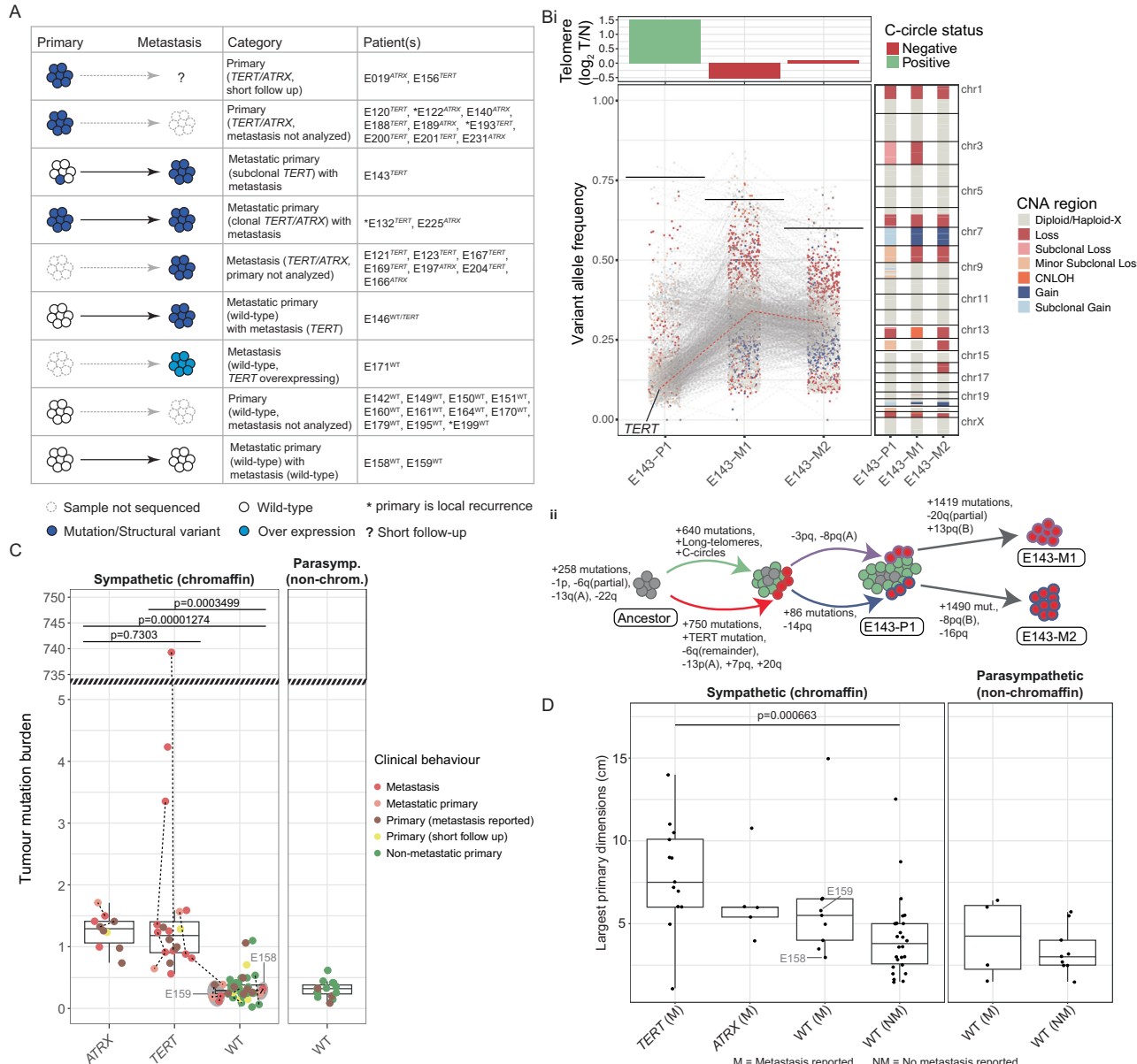

**Fig. 6 | *TERT/ATRX*-alterations and their association with metastatic progression. A** A schematic of data availability and *TERT/ATRX* status in primary tumours reported as metastatic. **B** A metastatic C-circle positive and *TERT*-mutant polyclonal primary: (i) The lower-left panel shows the variant allele frequency (VAF, y-axis) for all somatic mutations in the metastatic primary (E143-P1) and paired metastases (E143-M1/2) (x-axis). Copy number status in the region of each variant is indicated by dot colour. Horizontal lines indicate tumour purity as a proportion (read from y-axis). Grey lines connect shared mutations while a p*TERT* mutation is highlighted with a red line. The top panel shows the tumour/normal telomere content ratio, the colour of each bar indicates C-circle status. The panel on the right shows the genomic copy number status along each chromosome (y-axis) for each tumour (x-axis). (ii) A schematic illustration of the clonal evolution of metastatic disease in patient E143. Cell colour indicates the presence of the ALT phenotype (green) or p*TERT* mutation (red). **C** The

tumour mutation burden (mutations per megabase, y-axis) observed in each tumour with respect to *ATRX*^mut (*n* = 10), *TERT*^mut (*n* = 20) and wild type (*n* = {Chromaffin: 48, Non-chromaffin: 14}) status (x-axis). A one-tailed Student's t-test was used to test for statistical significance. The y-axis has been truncated to accommodate an extreme outlier which was excluded during statistical testing. **D** The dimensions (centimetres, y-axis) of the largest primary tumour reported for each patient. Patients are stratified by the presence of a *TERT/ATRX* mutation and the presence (M) or absence (NM) of metastatic disease (*n* = {Chromaffin *TERT*(M): 13, *ATRX*(M): 5, WT(M): 9, WT(NM): 26; Non-chromaffin: WT(M): 4, WT(NM): 9}). The hinges of each boxplot correspond to the first and third quartiles and the median value is marked. The whiskers extend to the largest and smallest value no greater than 1.5 times the interquartile range above or below the upper and lower hinges, respectively. A one-tailed Student's t-test was used to test for statistical significance.

1) E146 where the primary was *TERT* wild-type but the paired metastasis was p*TERT* mutant and 2) case E143 where a subclonal p*TERT* mutation was detected in the primary (VAF = 10.3%) but higher VAF was detected in two paired metastases (E143-M1 VAF = 33%, E143-M2 VAF = 28%), indicating all metastatic cells were p*TERT* mutant (Fig. 6B). Interestingly, the subclonal p*TERT* mutant primary (E143) also had detectable C-circles and longer telomeres, but these ALT features were absent in both metastases. The data therefore suggests the p*TERT* mutation and ALT features had

co-existed but were likely within different cell populations, while only the p*TERT* mutant cells metastasized.

Ten primary tumours from patients with metastatic disease did not have *TERT/ATRX* alterations, but the late acquisition of *TERT/ATRX* alterations and presence in metastatic cells could not be excluded given that the metastatic tissue was not sequenced. However, metastasis in the absence of *TERT/ATRX* alterations was also evident in three metastatic tumours for two patients (E158, E159). These metastases had no other

evidence of a TLM (i.e., they did not overexpress *TERT*, and had neither detectable C-circles nor an increased telomere length ratio) and had no recurrent cancer gene alterations that may explain metastatic progression. Interestingly, we found the tumour mutation burden was significantly elevated in *TERT/ATRX*-altered PCPG (one-sided Student's t-test p < 0.001) consistent with a previous study[11], but metastases from the *TERT/ATRX* wild-type cases had a low mutation burden (Fig. 6C). Interestingly, while it is has been previously shown that a larger primary tumour is associated with metastatic risk and *TERT/ATRX* mutations[11], only patients with *TERT* altered tumours in our series had significantly larger primaries when contrasting to patients without *TERT/ATRX* alterations (two-sided students t-test, *p* < 0.05)(Fig. 6D). Finally, survival analysis showed a significant association with poorer outcome in the *TERT/ATRX* altered group, which is also consistent with prior studies[11,15] (Supplementary Table 1, Supplementary Fig. 16).

## Transcriptional patterns in *TERT/ATRX* altered and metastatic PCPG

We next assessed transcriptional changes in *TERT* and *ATRX* altered PCPG, respectively, by contrasting each group independently to non-metastatic PCPG and directly to each other. Furthermore, we contrasted all metastatic and non-metastatic PCPG in a separate analysis (Fig. 7A). All HN-PG/non-chromaffin tumours (defined by transcriptional profile) were excluded given the absence of *TERT/ATRX* alterations in these tumours. Furthermore, we excluded low tumour purity cases (< 50% estimated by WGS data) and a subclonal *TERT*-mutant case. Contrasting the *TERT* and *ATRX* altered tumours to non-metastatic tumours we found 1152 and 1448 differentially expressed genes, respectively (Supplementary Data 6). Some of these genes were common to both analyses (*n* = 273), as well as when contrasting all metastatic and non-metastatic tumours (*n* = 180).

Consistent with prior studies, genes overexpressed in metastatic PCPG were enriched for cell cycle and proliferation gene ontologies[11,22]. *MKI67* was among the top overexpressed genes in keeping with elevated Ki67 IHC staining (Fig. 7B). Other genes included cell cycle regulators (e.g., *TOP2A*, *BUB1*, *FOXM1*, *AURKB2*, *CDK1*), that were highly correlated with *MKI67* expression (r > 0.88). Other genes overexpressed in metastatic PCPG included the polycomb repressor complex 2 gene *EZH2*, as previously described[22], and the transcription factors *OTX1* and *TPX1*.

Long non-coding RNAs and pseudogenes were enriched in genes differentially expressed in *ATRX* altered tumours (Pearson's Chi-squared *p*-value = 9.9e-14)(Supplementary Fig. 17). The serine/threonine protein kinase *RIPK4* was also overexpressed, while genes downregulated in *ATRX* altered tumours included the p53-dependent G2M cell cycle regulator *RPRM* and the GTP-binding gene *DRG2* – the latter gene also implicated in G2M cell cycle checkpoint control (Fig. 7C)[42]. Other genes down-regulated in *ATRX* altered tumours included *USE1*, *SULT4A1* and the nuclear-encoded mitochondrial gene *COX17*. Many differentially expressed genes were also within differentially DNA methylated regions (Fig. 7C). Gene-expression in neoplastic cells was confirmed using the snRNA-seq data and differential expression in *ATRX* mutant tumours was validated using independent bulk RNA-seq datasets[13,14] (Fig. 7D). Meanwhile, few genes were uniquely expressed in *TERT* altered tumours; amongst these were *IRX3*, *SDK1* and *TRIP13* (Fig. 7E).

Differential expression of small-RNA from *TERT* altered and *ATRX* altered tumours with non-metastatic primary tumours revealed down-regulation (adj. p < 0.05, log fold-change < −1.5) of four (hsa-miR-485-5p, hsa-miR-433-3p, hsa-miR-539-3p, hsa-miR-127-5p) and three (hsa-miR-3065-3p, hsa-miR-3065-5p, hsa-miR-148a-3p) miRNAs, respectively. In concordance with a previous study[43], three miRNA (hsa-miR-96-5p, hsa-miR-183-5p, hsa-miR-182-5p) were found to be upregulated (adj. *p* < 0.05, log fold-change > 1.5) in both *TERT* altered and *ATRX* altered tumours relative to non-metastatic primaries. These three miRNAs are collocated on chr7q32.2 and share targets including

Forkhead Box O1 transcription factor (*FOXO1*), snail family transcriptional repressor 2 (*SNAI2*), and RECK, a membrane-anchored matrix metalloproteinase regulator and potential inhibitor of metastasis[44–46] (Supplementary Fig. 18, Supplementary Data 7).

To explore which transcription factors may be driving changes in transcriptional programming in *TERT* and *ATRX* altered tumours we performed binding motif enrichment analysis using single-nuclei ATAC-sequencing. When compared to normal chromaffin cells, *TERT* altered cells demonstrated an enrichment of binding sites associated with the Activator Protein-1 (AP-1) transcription factor family (JUNB, JUND, FOSL1, FOSL2), as well as the SWI/SNF related BAF chromatin remodelling complex subunit C1 (SMARCC1), and the Wilms' tumour 1 (WT1) transcription factor. *ATRX*-mutant cells also showed an enrichment of peaks containing WT1 binding motifs, as well as the Sp1 and Sp2 transcription factors, however the greatest enrichment was seen for CTCF binding motifs. Interestingly, normal chromaffin cells showed an enrichment (i.e. depleted in *TERT/ATRX* altered cells) of the glucocorticoid (*NR3C1*) and androgen (*AR*) receptors (Supplementary Fig. 19).

## Treatment-related mutagenesis and resistance to DNA alkylating chemotherapy

DNA alkylating chemotherapies (e.g., dacarbazine and temozolomide) are common treatments for metastatic PCPG[47]. Two patients had paired tumours taken before and after cyclophosphamide, vincristine and dacarbazine (CVD) treatment. Case E169 had a para-aortic abdominal-thoracic PGL with distant lung, liver, and bone metastases (Fig. 8A). Tumours resected before (lung, E169-M1) and after (spinal disease, E169-M2) 23 cycles of CVD were used for genomic analysis. Paired E169 tumours had a high number of shared SNVs (*n* = 3134) including mutations in *TERT* and *BCOR*; however, the number of SNVs doubled in the post-CVD (*n* = 12607) tumour including mutations in *TP53*, *RPL5* and *POLE* (DNA Polymerase Epsilon) (Fig. 8B). SBS signatures also changed in the post-treatment sample with increased representation of SBS8, SBS12, SBS16 and SBS21 as well as indel signature ID1 (Fig. 8C). However, elevated SBS11 associated with DNA alkylating chemotherapy and SBS14 associated with DNA polymerase epsilon-deficiency was absent. Importantly, *MGMT* was overexpressed in the post-treatment tumour (E169-M2) (Fig. 8D). *MGMT* over-expression is a known temozolomide resistance mechanism in glioblastoma cells[48] and therefore potentially an acquired dacarbazine resistance mechanism in PCPG.

A second case (E167) had a left adrenal primary PC and then 14 years later developed HN-PGL and metastatic disease in the hip (Fig. 8E). The patient progressed with development of liver metastases after [131]I-MIBG therapy and was subsequently treated with 22 cycles of CVD. Further progression occurred with new intracranial and osseous disease. Liver and intracranial metastases represented disease pre- and post-CVD treatment respectively, with clonal relatedness confirmed by common SNVs (*n* = 2404) including a pTERT mutation (Fig. 8F). *MGMT* expression was low in both tumours (Fig. 8D). The post-CVD metastasis acquired more than a million additional SNVs corresponding to a strong SBS11 signature (Fig. 8G). Interestingly, the post-treatment intracranial metastasis (E167-M2) had a somatic *NRAS* (NP_002515.1:p.G12A) mutation and predicted to have elevated MAP kinase pathway activity by WTS (Supplementary Fig. 20). Importantly, a protein truncating *MLH1* splice-site mutation (chr3:g.37091976 G > A) was detected in the post-CVD metastasis with concordant loss-of-heterozygosity and low *MLH1* expression (Fig. 8F, H). Canonical SBS mismatch-repair (MMR) mutational signatures (e.g. SBS6, SBS14, SBS21) were not detected in the post-CVD treated tumour; however, insertion/deletion signature ID2 was elevated and insertion/deletion events at microsatellite sites revealed a high MSI-score (11.7 microsatellite indels per megabase) in the post-CVD tumour (E167-M2), which is above the prescribed microsatellite instability threshold ( > 4)(Supplementary Fig. 21).

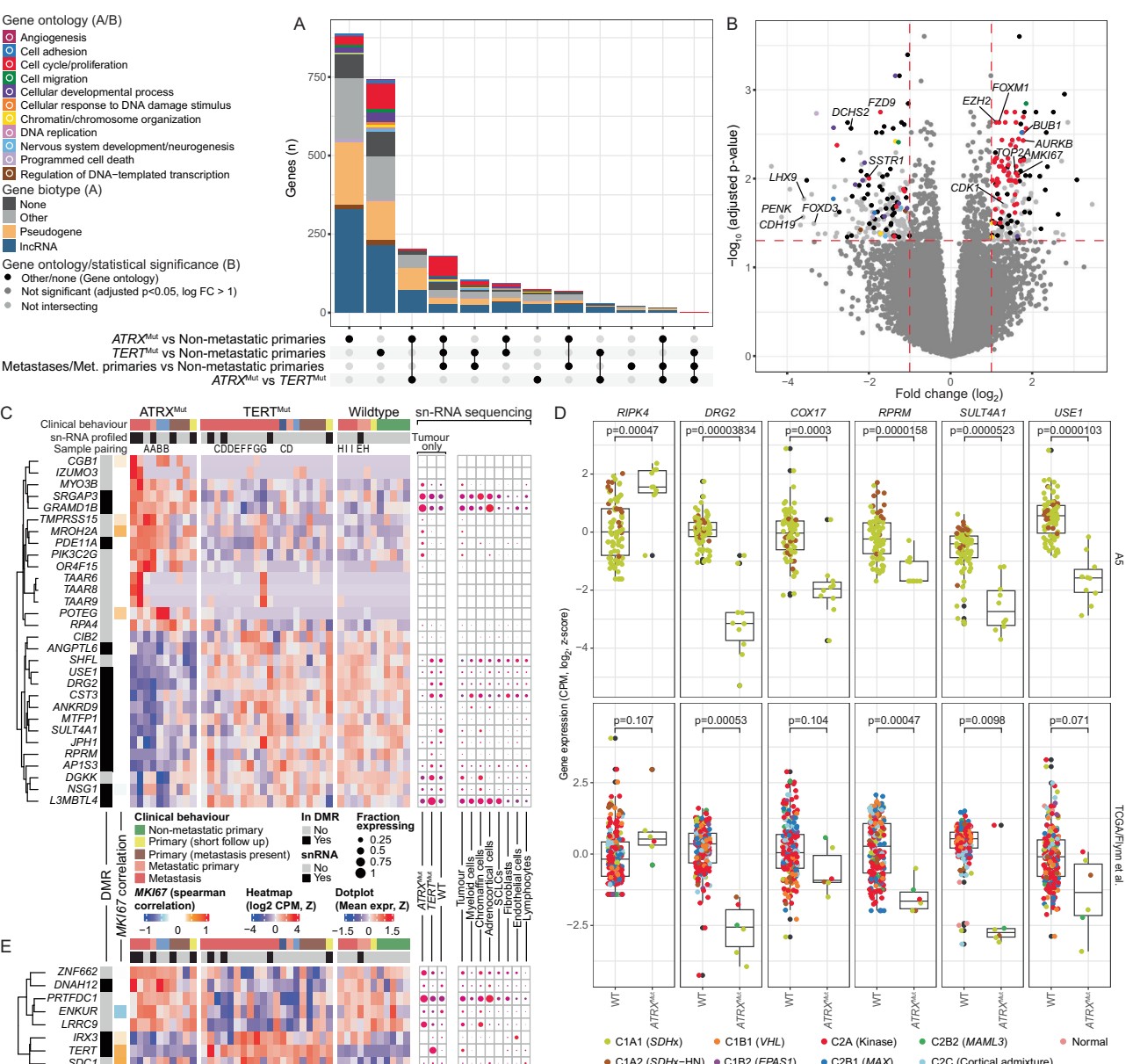

**Fig. 7 | Differential gene expression between _TERT_ and _ATRX_-altered and non-metastatic tumours.** Differential gene expression contrasting non-metastatic primary tumours with either (i) _ATRX_ mutant tumours, (ii) _TERT_ mutant tumours, or (iii) all metastases and metastatic primaries. A fourth contrast between _TERT_ and _ATRX_ mutant tumours was also performed. **A** An upset plot showing the intersection of genes that were significant (limma moderated t-test Benjamini-Hochberg adjusted _p_-value < 0.05, log-fold-change > 1) in each contrast. Bar colour indicates gene-ontology association for protein coding genes or gene biotype for non-protein-coding genes. **B** Differential gene-expression between non-metastatic primary and metastatic tumours showing fold change (log₂, y-axis) versus _p_-value (-log₁₀, x-axis, limma moderated t-test Benjamini-Hochberg adjusted). Genes that were also significant in non-metastatic primary vs _TERT_ altered and non-metastatic primary versus _ATRX_ altered contrasts are coloured according to gene ontology annotation. **C** Heatmap (centre panel) showing genes differentially expressed in both the _ATRX_mut vs non-metastatic primary and _ATRX_mut vs _TERT_mut contrasts (n = {_ATRX_mut: 9, _TERT_mut: 15, non-metastatic primary: 21}). Annotation bars (left) indicate whether the gene was found in a differentially methylated region (_ATRX_mut vs non-

metastatic primary), and the correlation of the expression of each gene to that of _MKI67_ (Spearman correlation). The right panel shows snRNA-seq expression aggregated by cell type (right sub-panel) or _ATRX/TERT_ mutation status (left sub-panel). Dot colour indicates mean expression while dot size indicates the fraction of cells expressing the gene (nCells = {_ATRX_mut: 29533, _TERT_mut: 19177, Tumour-wild-type: 14916; Tumour: 63626, Myeloid cells: 2198, Chromaffin cells: 3238, Adreno-cortical cells: 3723, Schwann-cell-like cells (SCLCs): 727, Fibroblasts: 1855, Endo-thelial cells: 1849, Lymphocytes: 1200}. **D** Expression (log₂ CPM, y-axis) of genes differentially expressed between _ATRX_ altered and non-metastatic primary tumours. Expression data is shown from the A5 cohort (top) and the TCGA/Flynn et al. cohorts (bottom). Data point colour indicates PCPG subtype. A two-sided Student's t-test was used to test for differences (n = {A5 - WT: 81, _ATRX_mut: 10, TCGA/Flynn - WT: 217, _ATRX_mut: 6}). **E** Differentially expressed genes in both the _TERT_-altered vs non-metastatic primary and _ATRX_mut vs _TERT_mut contrasts (n = {_ATRX_mut: 9, _TERT_mut: 15, non-metastatic primary: 21}). See panel (**C**) description for panel elements. Box plots to be interpreted as per Fig. 6.

Pathogenic variants in MMR genes (_MLH1, MSH2, MSH6, PMS2_) as well as treatment-related hypermutation and SBS11 have been described in temozolomide-treated glioblastomas[49,50] and more recently also pancreatic neuroendocrine tumours[51]. To search for evidence of MMR-deficiency in additional PCPG we analysed variant data from AACR

Project GENIE (release 13), representing comprehensive cancer panel data for 75 PCPG. We identified one hypermutated case with a dominant C > T mutational pattern and an _MSH6_ variant predicted to be a driver by Cancer Genome Interpreter (GENIE-DFCI-001077-11300, NP_000170.1:p.G162E) (Fig. 8I, J). Furthermore, we performed 523-

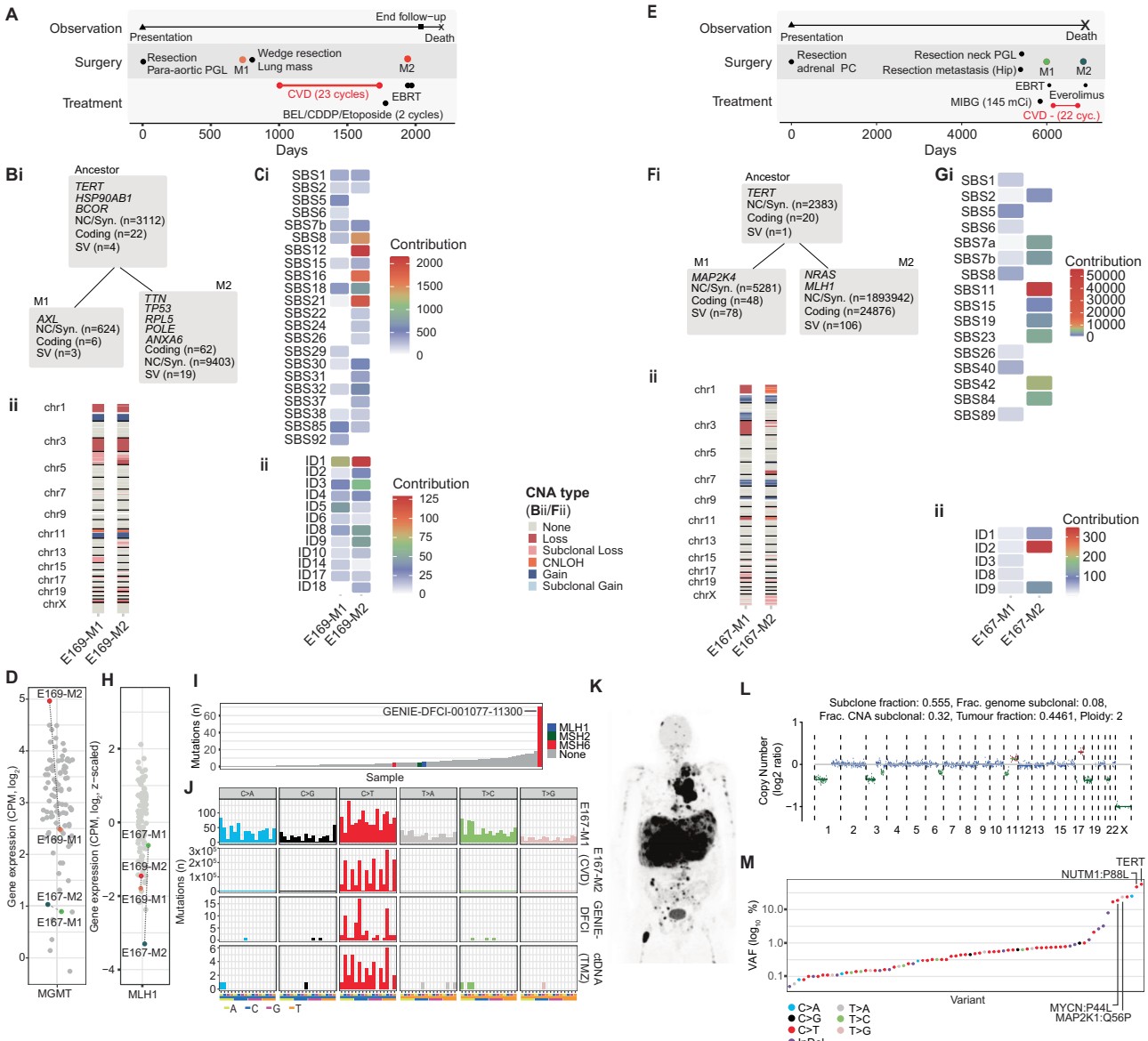

**Fig. 8 | Evolution of PCPG under treatment pressure. A** Clinical timeline of for patient E169. **Bi** Shared and private coding, non-coding/synonymous (NC/Syn.), and structural (SV) variants between paired metastases taken before and after CVD treatment. **Bii** Copy number status along each chromosome (y-axis) in paired metastases (x-axis). **C** Mutation signature analysis using COSMIC v3 SBS (i) and InDel (ii) signatures (y-axis) in paired metastases (x-axis). Heatmap colour indicates signature contribution. **D** Expression of *MGMT* in the A5 cohort. **E**–**G** Patient E167, see description for panels (**B**, **C**). **H** Expression of *MLH1* in the A5 cohort. **I** Total mutation counts (y-axis) for PCPG tumours (x-axis) in the Project GENIE data registry. Bar colour indicates the presence of a mutation in the mismatch repair gene

pathway. **J** Trinucleotide context for mutations observed in E167-M1 (top), E167-M2 (second from top), the highest mutation load tumour from the GENIE dataset (second from bottom), and ctDNA derived from a patient with metastatic SHDB-related PGL treated with Temozolomide. **K** ¹⁸F-FDG-PET imaging for Temozolomide treated patient at time of blood draw for cfDNA analysis. **L** IchorCNA analysis of ctDNA derived from Temozolomide treated patient (**M**) Variant allele frequencies (y-axis) for somatic variants (x-axis) observed in ctDNA derived from Temozolomide treated patient. Datapoints are coloured to indicate transition/transversion or insertion/deletion type.

gene panel sequencing of plasma cell-free (cf) DNA from a patient with *SDHB*-associated PCPG with high disease volume as shown by ¹⁸F-FDG-PET (Fig. 8K). Blood was taken after three cycles of temozolomide combined with ¹⁷⁷Lutetium-DOTA-octreotate. The patient had high fraction of circulating tumour (ct) DNA (45%) (Fig. 8L). Hypermutation was evident in cfDNA with a predominance of C > T transitions corresponding to SBS11 (Fig. 8J), although most C > T mutations had low VAF (~ 0.5%) in contrast to truncal mutations involving p*TERT, MYCN, NUTM1* and *MAP2K1* (VAF > 10%) (Fig. 8M). A mutation in an MMR gene was not detected; however, the hypermutated clone likely represented a small fraction of total disease volume therefore detection of an MMR

gene mutation may have fallen below the detection limit of the cfDNA assay (VAF > 0.5%).

## Discussion

We performed a comprehensive genomic analysis of *SDHB*-associated PCPG thereby creating a data resource to understand development of metastatic disease in these patients. Consistent with prior studies, most PCPG had stable genomes with few recurrently mutated cancer genes[11,13,14,20]. Importantly, we confirm *SDHB* mutant PCPG are molecularly distinguishable from other PCPG genotypes, while para-sympathetic (non-chromaffin) tumours including HN-PG had a very

distinct molecular profile. Our data confirms *TERT* and *ATRX* alterations are the most common secondary driver events in PCPG and are enriched in metastatic disease[11,13,15]. However, a small number of metastatic PCPG lack *TERT/ATRX* alterations, suggesting yet unidentified mechanisms of metastatic progression in these cases.

Mutual exclusivity of *TERT/ATRX* alterations in PCPG suggests these events have a common and potentially redundant role in metastasis. Both genes are involved in telomere maintenance - a hallmark of cancer[52]. However, given the majority of PCPG, including a subset of metastatic PCPG, have no obvious telomere lengthening mechanism (TLM), how, or indeed whether, these tumours maintain their telomeres is unknown. While methods such as the Telomere Repeat Amplification Protocol (TRAP)[53] and the C-circle assay are required to exclude the presence of a known TLM in tumour cells, absence of TLM has also been reported in other neuroepithelial cancers[54,55]. Meanwhile, while *ATRX/TERT* alterations were strongly associated with metastatic progression, the presence of long telomeres in the absence of TLM appears unlikely to be sufficient to drive metastatic progression[15].

ATRX is a SWI-SNF chromatin remodelling protein with multiple functions beyond telomeric regulation, while TERT is also thought to have non-canonical functions[56]. We observed that increased cell cycle activity appears to be common to *ATRX/TERT* altered PCPG, which is consistent with prior studies[11,22] and in keeping with increased Ki67 staining within metastatic PCPG[57]. Experimental models implicate *ATRX* and *TERT* in regulating cell cycle activity. For instance, ATRX binds to regulatory elements of cell cycle genes including *CHEK1*, as previously reported in a glioblastoma model[58]. Intriguingly, we found repression of *RPRM* and *DRG2* in *ATRX* altered PCPG, which are known to regulate the G2M cell cycle checkpoint[42,59]. It remains to be determined if *RPRM* and *DRG2* are ATRX target genes. TERT can also promote cellular proliferation and has a pro-survival function. For instance, cellular proliferation can be inhibited by wild-type reversion of a p*TERT* mutation in a glioblastoma cell line[60]. Furthermore, catalytically inactive TERT can rescue TERT-null cells, indicating a pro-survival function that is independent of telomerase activity[61]. How TERT promotes cellular proliferation is poorly understood and may be cell type and context dependent. Prior studies have shown TERT occupancy at gene-promoters including tRNAs[62] while TERT has also been shown to interact with oncogenic pathways involving MYC, WNT/B-catenin or NFkB[56]. Given few differentially expressed genes were restricted to *TERT* altered PCPG, experimental models will be required to understand any non-canonical TERT functions in PCPG.

*ATRX/TERT* alterations have potential implications for treatment and surveillance. An increased sensitivity of *ATRX* altered tumour cells to DNA damage response (DDR) inhibitors targeting ATM[58], WEE1[63], ATR and PARP[64,65] has been reported in other neuroectodermal cancers, and therefore might be effective in PCPG. Sensitivity to radio-sensitizing DDR-targeting drugs in combination with radionuclide therapies (e.g., [177]Lutetium-DOTA-octreotate or [131]I-MIBG) also warrants investigation in the context of *ATRX/TERT* alterations[66,67]. Although we confirmed *TERT/ATRX* alterations are associated with metastatic disease, the utility of these mutations as predictive biomarkers may be limited by the late timing of these events. Detection of *TERT/ATRX* alterations by cfDNA analysis may be a viable alternative to tissue biopsies, although the broader utility of cfDNA detection in PCPG still requires further validation.

DNA alkylating agents are commonly used for PCPG but there are currently no validated biomarkers to predict treatment response. We identified hypermutation together with MMR mutations (e.g., *MLH1*, *MSH6*) or *MGMT* overexpression in dacarbazine or temozolomide-treated PCPG. These observations are potentially clinically important. Firstly, monitoring PCPG patients for development of MMR mutations, SBS11 or potentially *MGMT* overexpression may indicate development of treatment-resistant cells. Secondly, MMR mutations may sensitize

tumour cells to other treatments. For instance, hypermutation can theoretically generate neoantigens causing T-cell recognition in tumours, making them more immunogenic and potentially more sensitive to immune checkpoint inhibition (ICI), although this theory is tempered by observations in glioblastoma where temozolomide-induced hypermutation did not translate to ICI-response[50]. Synthetic lethality could be another approach in MMR-deficient tumours, including use of WRN helicase inhibitors[68]. Further investigation is required to determine whether these treatments are efficacious in MMR-deficient PCPG.

Although therapeutically actionable mutations are rare in *SDHB* mutant PCPG, co-operative drivers can still represent potential treatment opportunities in some patients. In particular, somatic *EPAS1* variants in *SDHB*-mutant tumours may have translational relevance. It is currently unknown whether HIF2α inhibitors (e.g., belzutifan) are effective for *SDHB* mutant PCPG, but it is plausible that co-operative *EPAS1* mutations could make these tumours more sensitive[69]. Screening for co-operative somatic *EPAS1* mutations in *SDHB*-mutation carriers should therefore be employed for future HIF2α inhibitor clinical trials. Although we have reported a comprehensive analysis of *SDHB* mutant PCPG, future studies will be needed to validate our findings and identify additional low frequency genomic features that may be used as clinical biomarkers and future treatment targets.

## Methods

### Patient samples and human research ethics

The research was conducted under an approved protocol of the human research ethics committee at Peter MacCallum Cancer Centre under the guidelines of the National Health and Medical Research Council and in accordance with the Helsinki Declaration of 1975, as revised in 1983. Patients were recruited at 11 sites under protocols approved by their respective institutional review boards with written informed consent without any compensation. Patient recruitment sites included those forming the American-Australian-Asian-Adrenal-Alliance (A5) International Research Consortium including the Peter MacCallum Cancer Centre (Australia), the Kolling Institute Neuroendocrine Tumour Bank under a protocol approved at North Sydney Local Health District (Australia), National Institute of Health (USA), University of Colorado (USA), University of Texas Health Science Center at San Antonio (USA), University of Florida (USA), University of Michigan (USA), Centre hospitalier de l'université de Montréal (Canada), as well as four non-A5 sites at Tufts Medical Centre (USA), Waikato Hospital (New Zealand), the National Cancer Centre (Singapore), and Uppsala University (Sweden). Patient sex was determined by genomic profiling. The findings of this study apply equally to both sexes.

### Tissue sample review and nucleic acid extraction

Tissue samples were assessed for tumour cell content by a single pathologist (AG) by haematoxylin and eosin staining of 4uM cryosections. Ten serial 10uM sections were taken for nucleic acid extraction adjacent to the H&E reviewed sections. If required, microdissection of tissue was performed on ethanol fixed and stained fresh tissue sections to dissect tumour regions from the sectioned material. DNA was extracted from tissue samples and matched patient whole blood using the DNeasy® Blood & Tissue kit (Qiagen, Germany) according to manufacturer's protocol. Total RNA was extracted from tissue using the miRNeasy mini kit (Qiagen, Germany) according to the manufacturer's protocol. Nucleic acids were quality control checked by Agilent TapeStation microfluidic electrophoresis (Agilent, USA) and NanoDrop spectrophotometer (Thermo Fisher Scientific, USA).

### Targeted amplicon sequencing

To validate mutations of interest, targeted amplicon sequencing was done using custom primers compatible with universal indexed primers for Illumina cluster generation. PCR amplification for the region of

interest was done in a single well (for each patient sample) and the expected PCR amplicon product then verified by agarose gel electrophoresis. Multiple amplicons for each patient sample were pooled, purified, and then barcoded with the index primers, quantified by fluorometer (Qubit dsDNA High-sensitivity, USA) and then sequenced on a MiSeq using a V3 300 cycle kit (Illumina, USA). FASTQ files were aligned to human genome (GRCh38) using BWA-mem and BAM files were visualized in IGV[70] to confirm variant of interest and record the variant allele frequency.

### Whole genome sequencing

**Library preparation.** Libraries were prepared using the Illumina® TruSeq™ DNA Nano library preparation method according to the manufacturer's instructions at The University of Melbourne Centre for Cancer Research (UMCCR) using 200 ng input DNA and a 550 base pair insert size. Samples were sequenced in separate batches on the Illumina® Nova-Seq 6000 according to manufacturer's instructions (Illumina, USA).

**WGS alignment and small variant calling.** BCL files were demultiplexed and converted to FASTQ files with Illumina® bcl2fastq (version 2.20.0.422) with default settings (including adapter trimming). Alignment and variant calling steps were run using the bcbio-nextgen cancer somatic variant calling pipeline (version 1.2.4-76d5c4ba) (https://github.com/bcbio/bcbio-nextgen). The versions of all programs used by bcbio-nextgen are additionally shown in Supplementary Table 2. Briefly, the tumour and normal sequencing data in FASTQ format were first processed by Atropos (version 1.1.28)[71] to clip homopolymers (minimum 8 bases) from the ends of reads. Trimmed FASTQ files were aligned to the human genome (GRCh38) with BWA-mem (version 0.7.17)[72] with predominantly default settings. The two exceptions were that the -M flag was enabled to mark shorter split read hits as a secondary alignment and seeds with >250 occurrences were skipped (reduced from the default 500).

Germline variants were called by vardict (version 1.8.2)[73], GATK HaplotypeCaller (GATK version 4.1.8.1)[74,75] and Strelka2 (version 2.9.10)[76] variant callers. Somatic variants were called by vardict, Mutect2 (see GATK version)[77] and Strelka2 (version 2.9.10) variant callers. In both instances a variant was accepted as true if it was detected by 2 of the 3 callers. Somatic variants were further post-processed by umccrise (version 1.2.0-rc.1, https://github.com/umccr/umccrise). To further reduce spurious variant calls, two variant blacklists were created. The first was generated by counting supporting reads for all detected somatic variants across all normal BAM files using SAMtools[78] (version 1.12) mpileup. Any somatic variants that were supported by three or more reads in more than two normal controls were excluded. The second blacklist was created by extracting all observations of pass-filter somatic variants from unfiltered variant call files (i.e., including those that were rejected by the caller) for all three somatic callers. The number of observations of a given variant across all samples and callers were grouped by pass or reject status and summed. Any variant with a reject to pass ratio of greater than one was placed on the blacklist.

**WGS copy number and structural variant calling.** Structural variants (SV) were called with GRIDSS (version 2.8.0)[79]. GRIDSS output was annotated for repeat sequences using RepeatMasker (version 4.1.5) and filtered for somatic SVs using GRIPSS (version 1.11). To ensure high quality somatic calls, all SVs passing GRIPSS filtering were examined in the raw output of GRIDSS across all samples. Any SV (based on exact genomic coordinates) that was observed (whether ultimately rejected or passed) in more than two unrelated samples was removed. Finally, SVs were required to have either a minimum of three supporting split reads or paired reads, or a combination of at least one split and one paired read. Somatic copy number profiles were generated using PURPLE (version 3.1, https://github.com/hartwigmedical/hmftools) from input generated by COBALT (version 1.11), AMBER (version 3.5), and GRIDSS. Structural variants were further annotated with Linx (version 1.16) using the output from PURPLE.

**Telomere length estimation and telomere variant repeat detection.** Telomere content was estimated from WGS for each tumour and normal pair using TelomereHunter (version 1.1.0)[80] with default parameters. Telomere content was computed from the number of intra-telomeric reads normalized by the total number of reads with a GC composition like that of telomeres. Telomeric enrichment in tumours was defined as a $\log_2$ ratio of tumour telomere content to blood telomere content greater than 0.5. TelomereHunter was also applied to WTS for each tumour to estimate TERRA content.

**Mutational signature analysis.** For each sample, the single base substitution (SBS), doublet base substitution (DBS), and small insertions and deletions (ID) mutation count matrices were extracted from variant call format (VCF) files containing somatic variant calls using the MutationalPatterns (version 3.10.0)[81] package for R. The optimal linear combination of the COSMIC V3[82] SBS, DBS, and ID mutational signatures that best reconstructed their respective profiles was then estimated using the fit_to_signatures function.

**Microsatellite instability assessment.** Insertions and deletions in microsatellite regions were assessed using PURPLE (version 3.1). Samples with more than four microsatellite Indels per megabase were considered to have microsatellite instability as per the software default setting.

**Identification of recurrent copy number alterations.** The Genomic Identification of Significant Targets in Cancer (GISTIC, version 2.0)[83] algorithm was used to identify recurrent copy alterations. The input data for GISTIC was modified from the somatic segmentation output from PURPLE by taking $\log_2$ of the segmental copy number incremented by a small offset (0.01) and subtracting one to centre the diploid state around zero ($\log_2(CN + 0.01)$-1). GISTIC was run in broad mode with segments containing less than 50 markers undergoing joining. The broad length cutoff was set to 75% of a chromosome and arm-peel was activated. The analysis was applied independently to sample subsets including HN-PG/parasympathetic PG, non-metastatic and metastatic sympathetic PCPG. In cases where a patient had more than one primary tumour, one was chosen at random, in cases where a paired primary and metastasis were present, the metastasis was used.

**Identification of driver genes.** The dNdScv R package[33] was used to assess somatic mutations to detect genes under positive selection. Before running the analysis, quality filtered somatic variant calls were lifted over from the GRCh38 to HG19 genome build using the rtrack-layer package for R with a liftover chain obtained from UCSC (hg38ToHg19.over.chain). The resulting mutation set was analysed with dndscv function using default parameters. Additionally, the HG19 mutation set was used to run the OncodriveFML[34] driver detection tool via the web-based application interface (https://bbglab.irbbarcelona.org/oncodrivefml/home, accessed 5th February 2024). The algorithm was run using genomic region subsets encompassing coding sequences or promotor regions with Combined Annotation Dependent Depletion (CADD) scores version 1.0.

**Clonal evolution analysis.** For each patient with paired samples, variants detected in any sample from a given patient were recalled using SAMtools (version 1.12) mpileup in all samples for that patient. The resulting allele counts were then used as input for Pairtree (version

1.0.1)[84]. Variants that were detected in between zero and three reads in any sample were marked for exclusion from all samples for a given patient as it was difficult to discern sub-clonality from sequencing artifacts. Variant read probabilities (VRP) were computed based on copy number segment data from PURPLE. In diploid regions the VRP was set to 0.5. In regions of single copy loss or copy-neutral loss of heterozygosity where the variant allele fraction was greater than half the sample purity the VRP was set to 1. In regions of copy-neutral loss of heterozygosity where the variant allele fraction was less than half the sample purity the VRP was set to 1 divided by the major allele copy number. For all other region types the VRP was set to 1 divided by the total copy number. Clustering was performed using the *clustervars* script. The resulting clusters were manually refined and integrated with copy number data to produce evolutionary trees.

## RNA-seq

**Library preparation.** Samples were prepared using the NEB-Next directional RNA-Seq kit (NEB, USA) and underwent 150 bp paired-end sequencing on the Illumina NovaSeq 6000 (Illumina, USA) according to manufacturer's instructions.

**Read quantification.** RNA sequencing reads were aligned with *STAR*[85] (version 2.7.9a) using a two pass approach. First, all data was aligned using a *STAR* reference built with the Gencode human reference genome with decoy HLA sequences (GRCh38) and Gencode gene transcript annotation file (version 36, primary assembly annotation). Following alignment, the splice junction output from all samples was merged. Supporting read evidence for each junction was summed and junctions with less than 10 unique supporting reads were removed. The resulting junction file was used to build a new *STAR* reference and the alignment was repeated. Gene level feature counts were obtained with *HTSeq-count* (version 0.11.3) using the previously described Gencode transcript annotation file. All parameters were default except Mode which was set to intersection-nonempty.

**Differential gene expression.** To contrast gene expression between parasympathetic and sympathetic (including pheochromocytoma) paraganglioma, samples were assigned to one of two groups based on clinical annotation and UMAP clustering. Samples where UMAP clustering did not reflect the clinical annotation or tumour purity was below 50% were excluded from the contrast. To perform differential expression analysis, transcript counts obtained from HTSeq-count were converted to a digital gene expression object using the *edgeR* (version 3.42.4)[86] package for R. Lowly expressed genes were removed using the *filterByExpr* function with default parameters. Library scaling factors were then computed using the *calcNormFactors* function. A design matrix was constructed without an intercept to include the differential group and patient sex as a covariate. Normalization factors were applied and correction weights for count-dependent variance were computed using the *voom* function from the *limma*[87] (version 3.56.2) package for R. To account for multiple samples from the same patient, the inter-duplicate correlation was computed using the *duplicateCorrelation* function with the patient identifier as the blocking factor. Linear model fitting was performed using the *lmFit* function supplied with the duplicate correlation computed above and the patient identifier as the blocking variable. Contrasts were fitted using the *contrasts.fit* function and test-statistics were computed using the *eBayes* function.

To contrast metastatic, *TERT*-altered, and *ATRX*-altered tumours with non-metastatic primary tumours samples were assigned to groups based on clinical behaviour. The clinical behaviour groups were non-metastatic primary tumour, primary tumour where metastatic disease was present, metastatic lesion, or a tumour with short clinical follow up. These groups were then further segregated based on the presence or absence of a *TERT* or *ATRX* mutation. Parasympathetic tumours and tumours with low purity (< 50%) were excluded from all

the contrasts. Samples with an *ATRX/TERT* alterations where the allele fraction was less than 25% were excluded from genotype-centric contrasts, and samples with short clinical follow-up were excluded from contrasts involving clinical behaviour. See Supplementary Data 2 for a full list of contrast participation. Differential expression was performed as described above.

**RNA fusion detection.** Whole transcriptome sequencing data aligned with STAR aligner (described above) was analysed with the *Arriba* (version 2.1.0) gene fusion detection tool using the supplied blacklist, known fusion, and protein domain files with default parameters.

**Harmonization with published RNA-seq datasets.** Read counts previously published by Fishbein et al.[13] were obtained from the Genomic Data Commons using the TCGAbiolinks[88] (version 2.29.1) package for R with these parameters: project = "TCGA-PCPG", data.category = "Transcriptome Profiling", data.type = "Gene Expression Quantification", workflow.type = "STAR - Counts". Additional data previously published by Flynn et al.[14] was obtained from the European Genome-Phenome Archive (EGAS00001005861). Raw counts from the publicly available data were merged with our dataset retaining genes present in all three datasets. Count matrices were converted to a digital gene expression object with *edgeR* and lowly expressed genes were removed using the *filterByExpr* function with default parameters. Library scaling factors were then computed using the *calcNormFactors* function and counts were transformed to log counts per million using the *cpm* function. Possible technical variance was ameliorated using the *removeBatchEffect* function from the *limma* package where the dataset identifier was supplied as the batch and the primary driver genotype (e.g., RET, SHDx, VHL) as the factor to preserve.

## Small RNA-seq

**Library preparation.** Samples were prepared using the NEXTFLEX® Small RNA-Seq Kit v3 (Bioo Scientific). Samples were sequenced at the molecular genomics core (Peter MacCallum Cancer Centre) using 50 bp single end sequencing on the Illumina NextSeq 500 (Illumina, USA).

**Read quantification.** FASTQ files were pre-processed with two rounds of *cutadapt* (version 1.6.3) to first remove adapter sequences (-a TGGAATTCTCGGGTGCCAAGG, --minimum-length 23) and then remove the first and last four bases (-u 4, -u -4). Following trimming, sequences were aligned to GRCh38 with *subread* (version 1.6.3) and quantified using *featurecounts* (version 1.6.3) in stranded mode against the miRBase (version 20) miRNA transcripts[89]. Following read quantification, read-count processing was performed in an identical manner to that described under RNA-Seq analysis.

**Harmonization with published small-RNA-seq datasets.** Small-RNA sequencing data previously published by Castro-Vega et al.[20] were obtained from ArrayExpress (E-MTAB-2833). Additional data published by Fishbein et al.[13] were downloaded from the Genomic Data Commons using the GDC Data Transfer Tool in BAM format and converted to FASTQ format using SAMtools (1.10). Alignment and read quantification were performed as described above under 'Read quantification'. Read count harmonization was performed as described for RNA-seq dataset under 'Harmonization with published RNA-seq datasets'.

**Differential expression of small-RNA dataset.** Differential expression analysis was performed as outlined for whole transcriptome RNA-sequencing. The same contrast groups were used, however, samples that were determined to be strong outliers by UMAP clustering of small-RNA expression data were excluded from the analysis. To ensure biological relevance for miRNA activity, genes were required to have at least 100 counts per million in more than one sample and a total of

more than 1000 counts per million across all samples to be included in the analysis.

## Methylation analysis

**Methylation array processing.** Methylation profiling was performed by the Australian Genome Research Facility (AGRF). Tumour DNA samples were normalized to approximately 200 ng of DNA in 45 μL. Bisulfite conversion was performed with the EZ-96 DNA Methylation MagPrep (Zymo Research, California, USA). Bisulfite converted DNA was processed on the Illumina Human Methylation EPIC BeadChip (Illumina, California, USA) following the Illumina Infinium HD Methylation assay protocol (AGRF NATA scope GGTMN00263).

**Methylation array data pre-processing and normalization.** The *minfi* (version 1.46.0) package for R was used to read in a targets manifest and red/green channel IDAT files using the *read.metharray.sheet* and *read.metharray.exp* functions, respectively. Probe detection $p$-values were determined with the *detectionP* function. Raw green/red values were converted to methylation signal and functionally normalized using the *preprocessFunnorm* function from *minfi*. Any probes with a detection $p$-value below 0.01 in any sample or those that fell on a SNP position (using *dropLociWithSnps*) were removed from the dataset. Furthermore, the *maxprobes* package for R was used to exclude any cross-reactive probes. Illumina probe annotation version 1.0 with GRCh38 coordinate information was obtained from: https://github.com/achilleasNP/IlluminaHumanMethylationEPICanno.ilm10b5.hg38.

**Probe level differential methylation analysis.** Probe level differential methylation analysis was performed using the Remove Unwanted Variation workflow with the RUV package via the *MissMethyl* (version 1.34.0) package for R[90,91]. Firstly, functionally normalized methylation values were converted to M-values using the *getM* function from the *minfi* package and negative control probes were obtained using the *MissMethyl*'s *getINCs* function. For each contrast, two rounds of variation adjustment were performed. In the first round, an RUV model was computed with RUV-inverse algorithm using the *RUVfit* function applied using the Illumina negative control probes as controls. The rescaled variances were then applied using the *RUVadj* function and $p$-values for differential methylation of each probe were obtained with the *topRUV* function. Probes with a $p$-value greater than 0.5 were selected for use as controls in the second round of variation adjustment using the RUVfit function and the resulting model was applied using the *RUVadj* function.

**Differentially methylated regions.** Identification of differentially methylated regions was performed with the DMRcate[92] package (version 2.14.1) for R. Firstly, functionally normalized methylation values were converted to M-values using the *getM* function from the *minfi* package. Contrast and design matrices were generated using the same process described for differential gene expression. For each contrast, individual probes were then annotated using the *cpg.annotate* function with analysis type set to differential and the default annotation set for Illumina EPIC arrays (ilm10b2.hg19). Differentially methylated regions were then identified using the *dmrcate* function with a lambda of 1000 and bandwidth scaling factor of 2. Significant DMRs were those with a Fisher multiple comparison statistic below 0.05.

**Harmonization with previously published methylation array datasets.** Illumina Infinium HumanMethylation450 methylation array data previously made available by Fishbein et al.[13] were obtained from the Genomic Data Commons using the TCGAbiolinks[88] (version 2.29.1) package for R with these parameters: project = "TCGA-PCPG", data.category = "DNA Methylation", data.format = "IDAT", legacy = False. The array data were processed as described under 'Methylation array data pre-processing and normalization' with the exception of the annotation

file, which was substituted with the supplied Illumina Infinium Human-Methylation450 annotation. Additional data made available by Letouzé et al.[9] were obtained from the Gene Expression Omnibus using the GEOquery package for R (accession GSE39198 and GSE43293). The GSE39198/GSE43293 data was only available as beta values and was not reprocessed. The beta-values from all three datasets were converted to m-values and merged, retaining only probes available on the Illumina Infinium HumanMethylation27 array design. A design matrix was generated with the *model.matrix* function from the limma package using the primary driver genotype (e.g., RET, SHDx, VHL, Unknown). Batch effects were then adjusted using the *removeBatchEffect* function from limma where the dataset was supplied as the first batch annotation and patient gender as the second batch annotation.

## Single nuclei RNA-sequencing

**Library preparation with 10x chromium.** snRNA-seq was performed using the 'Frankenstein' protocol (https://doi.org/10.17504/protocols.io.bqxymxpw). Briefly, nuclei were extracted from frozen tissues and subjected to fluorescence-activated nuclei sorting (FANS) using 4′,6-diamidino-2-phenylindole (DAPI). Sorting was performed using a BD FACSaria 2 instrument, sorting between 3000 and 10,000 nuclei per sample, capturing both diploid and tetraploid populations. FAN-sorted nuclei were immediately processed using either the 10x Chromium Single Cell 5′ (PN-1000006, 4 samples) or 3′ (PN-1000075, 4 samples) Library & Gel Bead Kit (10x Genomics, USA). Once processed, snRNA-seq libraries were sequenced on the Illumina Nova-Seq 6000 (Illumina, USA) using 150 bp paired-end sequencing.

**Data processing and analysis.** Additional snRNA-seq data for two normal adrenal medulla samples (E240, E243) previously published by Zethoven et al.[22] were obtained from the European-genome-phenome archive (accession EGAS00001005861). Single-nuclei RNA sequencing reads in FASTQ format were aligned and counted using *CellRanger* (version 7.2.0). Read-one data for libraries processed with the 3′ protocol was restricted to the first 26 bases using the r1-length flag. Feature counting was performed using the 10x supplied GRCh38 transcriptome (version refdata-gex-GRCh38-2020-A) with reads aligning to introns contributing to feature counts. Single-Cell Remover of Doublets (SCRUBLET, version 0.2.3) was used to identify gel-beads containing more than one cell. CellRanger count matrices were loaded using the *Read10X* and *CreateSeuratObject* functions from the Seurat (version 4.3.0.1) package for R in conjunction with the SCRUBLET output. For each sample, features present in less than three cells and cells containing less than 500 features or 750 unique RNA molecules, or more than 10 percent mitochondrial RNA were excluded. Cell cycle scoring was performed on normalized and scaled counts using the S-phase and G2M-phase marker gene lists provided with the Seurat package. The data was scaled again using *ScaleData* function with S-phase score, G2M-phase score, unique RNA molecule count, and percent mitochondrial RNA supplied as regression variables. Cell type clusters were found using the *FindClusters* function and cell types were assigned manually using marker genes for fibroblast (*FAP, PDGFRB, PDGFRA, ACTA2, COL1A1*), chromaffin (*PNMT, TH, DBH, CHGA, CHGB*), adrenocortical (*STAR, CYP11B1, CYP11A1*), endothelial (*FLT1, EPAS1*), Schwann-cell-like cells (*CDH19, SOX10, S100B, VIM*), monocyte/macrophage (*MSR1, CD163, CCL3*), and lymphocyte (*CD2, CD3E, MS4A1*) cell types.

## Single nuclei ATAC-sequencing

**Library preparation with 10x Chromium.** snATAC-seq was conducted using the "Van Helsing" protocol (https://doi.org/10.17504/protocols.io.bw52pg8e)[93]. Initially, nuclei were isolated following the steps outlined in the previously mentioned "Frankenstein" protocol (https://doi.org/10.17504/protocols.io.bqxymxpw)[94]. Sorted nuclei were collected into a round-bottom 96-well plate containing 100 μL of ice-cold Wash Buffer. The sorted nuclei were then transferred to 0.2 mL PCR tubes; 50 μL of

ATAC Wash Buffer-Dig was added to the well to ensure transfer of any remaining nuclei. The nuclei were then centrifuged at 500 x g for 5 min at 4 °C. The supernatant was carefully decanted, leaving behind approximately 10 μL. Without resuspending the pellet, 100 μL of ice-cold Diluted Nuclei Buffer was added. The sample was centrifuged again under the same conditions, and 100 μL of the supernatant was gently removed in two steps—first 90 μL, then the remaining 10 μL. Another 100 μL of Diluted Nuclei Buffer was added gently. This washing step was repeated, removing the supernatant in two steps to leave behind approximately 7–10 μL. The nuclei were resuspended in this small volume, ensuring thorough mixing by carefully washing the walls of the tube. For quantification, 1–2 μL of the nuclei suspension was diluted 1:5 with Diluted Nuclei Buffer and mixed 1:1 with Trypan Blue. Nuclei were counted using a Countess II FL Automated Cell Counter or a hematocytometer to estimate the expected number of nuclei based on the recovery factor. Finally, 5 μL of the nuclei suspension in Diluted Nuclei Buffer were used directly in the Chromium Single Cell ATAC Reagent Kits protocol (CG000168 Rev A), following the specific recommendations for low input samples to optimize the transposition reaction and subsequent library preparation. Once processed, snATAC-seq libraries were sequenced on the Illumina NextSeq 500 (Illumina, USA) using 50 bp paired-end, duel-indexed sequencing.

**Data preprocessing.** Raw sequencing data was demultiplexed using *cellranger-atac mkfastq* (V2.0.0). *Cellranger-atac count* was used to combine replicate sequencing runs, align single nuclei ATAC reads to the GRCh38 reference genome (GRCh38 2020 A 2.0.0, 10x Genomics), and call initial peak sets for individual samples. *Cellranger-atac aggr* was then used to combine samples into a single peak-barcode matrix.

**Quality control.** Signac[95] (version 1.12.9004) and Seurat (version 4.3.0.1) were used for quality control and downstream analysis of snATAC-seq data. Filters were applied to QC metrics that were independent of peaks, to avoid bias toward under-represented cell types. Transcription start site enrichment scores were calculated using the *Signac TSSEnrichment* function. Outlier nuclei with over 5000 unique reads were assumed to be doublets or nuclei clumps and removed. Nuclei with TSS enrichment scores below 2 were removed due to having a low signal-to-noise ratio.

**Cell type identification.** Cell type identification was performed via label transfer in Signac. A gene-by-cell accessibility matrix was computed by calculating the sum of reads aligning to the gene body and promoter regions of protein-coding genes for each cell. Log normalisation was performed on the gene activities with *Seurat NormalizeData*. Integration was performed to link snATAC-seq nuclei clusters to defined cell types within the snRNA-seq dataset generated from matched and similar samples[96]. Integration was performed with the *FindTransferAnchors* Seurat function. Briefly, Canonical Correlation Analysis (CCA) was used to identify shared sources of variation between the snATAC gene activity and snRNA gene expression, and project the two datasets into the same reduced-dimensional space. Mutual nearest neighbours (cell pairs from each dataset within each other's neighbourhood) were identified and used as integration anchors between the two datasets. These anchors were then used to transfer cell-type labels from the snRNA reference dataset to the snATAC-seq dataset, using the Seurat *TransferData* function, enabling prediction of snATAC-seq nuclei cell types. Cell type identification was confirmed by gene activity of canonical cell-type-specific marker genes.

**Peak calling and visualisation.** Open genomic regions were defined using pooled reads for each cell type and tumour sample with >100 total nuclei individually using the MACS2[97] peak caller (version 2.2.9.1) implemented in the Signac *CallPeaks* function. Resultant fragment pileup bedgraph files for each cell type were visualized using the SparK command line tool[98].

**Motif enrichment analysis from sn-RNA-seq data.** Motif analysis was performed using *ArchR* (version 1.02). Fragment counts output by *Cellranger-atac* were converted to arrow format using the *createArrowFiles* function (minTSS = 4, minFrags = 1000). Doublet scoring was performed using the *addDoubletScores* function. As some samples were reported to be homotypic, additional doublet detection was performed using AMULET (version 1.1). Doublets were then marked using a modified *filterDoublets* function which applied the standard ArchR filters and removed any cells considered to be doublets by AMULET. Cell type annotation was then transferred from the single nuclei RNA-seq data using the *addGeneIntegrationMatrix* function. A subset of the data containing only tumour and normal chromaffin cells was used to generate pseudo-bulk replicates using the *addGroupCoverages* (minRep = 3, maxRep = 10, minCells = 100, maxCells = 500, sampleRatio = 0.5) and *addReproduciblePeakSet* functions. The cell groups were defined as Tumour-*TERT*-mutant, Tumour-*ATRX*-mutant, and Normal-chromaffin-wild-type. Marker features were identified with the *getMarkerFeatures* function using the Wilcoxon test and applying bias adjustment for transcription start site enrichment scores and number of fragments in each cell. Enriched motifs in each contrast were determined using the *peakAnnoEnrichment* function with a false discovery rate cutoff of less than or equal to 0.05 and an absolute log fold-change greater than or equal to one.

### Cell-Free DNA panel sequencing

**Sample collection.** Patient blood was collected in Cell Free DNA BCT CE tubes (catalogue number 218996, Streck, USA). Within two hours of collection, whole blood was centrifuged at 1900 g (brake off) for ten minutes to separate the plasma. Plasma was aspirated and aliquots were centrifuged at 20000 g to pellet residual cells or debris. The plasma was stored at −80 °C until DNA extraction. Cell free DNA (cfDNA) was extracted from 5 mL of plasma using the Qiagen QIAamp circulating nucleic acid kit (catalogue number 55114, Qiagen, The Netherlands) according to manufacturer's instructions.

**Library preparation.** A sequencing library was generated with 30 ng of input cfDNA using the Illumina TruSightOncology 500 ctDNA library preparation kit (catalogue number 20039252, Illumina, USA). Input quantity was determined based on the mononucleosome fraction (75-300 bp) using the Agilent TapeStation4200. Library preparation was performed as per manufacturer's instructions with the addition of a 1.2x bead clean-up of the amplified enriched library. The fragment distribution of the enriched library was assessed on the TapeStation4200, and library quantification was performed using the KAPA Illumina library quantification kit (catalogue number 07960140001, Roche, Switzerland) and analysed with QuantStudio6 (ThermoFisher, USA).

**Sequencing and data processing.** Sequencing was performed at the University of Melbourne Centre for Cancer Research Centre using the Illumina NovaSeq 6000 (Illumina, USA) to achieve a depth of coverage greater than 35000x. Data was analysed with Illumina DRAGEN TruSight Oncology 500 ctDNA Analysis Software v1.2 on the Illumina Connected Analytics platform (ICA v1). PierianDx Clinical Genomics Workspace (v6.27.0.1) was used for variant interpretation.

### Immunohistochemistry
Immunohistochemistry was performed and interpreted on formalin fixed paraffin embedded tissue sections using methods that have been described in detail elsewhere[99,100]. Briefly, for both SDHB (ABCAM, Cambridge, UK: ab14714, clone 21A11, dilution 1:100) and Ki67 (clone M7240 Dako, Carpenteria, CA USA, dilution 1:50) commercially available mouse monoclonal antibodies were used.

## C-circle assay

Isothermal rolling circle amplification and detection of C-circles was performed as described[40,101]. For each tumour sample, 20 ng DNA was incubated with 10 μl 0.2 mg/ml bovine serum albumin, 0.1% Tween, 4 mM DTT, 1 mM each dATP, dGTP, dCTP and dTTP, 1× Φ29 Buffer, with or without Φ29 polymerase 7.5 U (New England Biolabs) at 30 °C for 8 hours, then at 65 °C for 20 min. Reaction products were diluted to 100 μl with 2× SSC and dot-blotted onto a 2× SSC-soaked Biodyne B nylon membrane (Pall Corporation). The membrane was air-dried, twice crosslinked using 1200 J UV (Stratagene) and hybridized overnight at 37 °C with end-labelled 32P-(CCCTAA) and PerfectHyb Plus hybridization buffer (Sigma). The membrane was then washed at 37 °C three times in 0.5x SSC/0.1% SDS and exposed to a storage phosphor screen, which was imaged using an Amersham Typhoon laser scanner (Cytiva) and analysed by densitometry using ImageQuant software (Molecular Dynamics). C-Circle levels for each tumour sample were reported as the difference in the dot-blot intensities resulting from reaction of the sample's DNA with and without Φ29 polymerase.

## Statistics & reproducibility

Student's t-tests and Pearson's Chi-squared tests for count data were performed using the *t.test* and *chisq.test* functions, respectively, from the base *stats* package for R (version 4.3.1). No statistical method was used to predetermine sample size. Samples and assays failing quality control measures for tumour content, nucleic acid quality, or assay specific controls were excluded. The experiments were not randomized. The investigators were not blinded to allocation during experiments and outcome assessment.

## Reporting summary

Further information on research design is available in the Nature Portfolio Reporting Summary linked to this article.

## Data availability

Raw FASTQ files for WGS, WTS, small-RNA-sequencing, sn-ATAC-sequencing, sn-RNA-sequencing, as well as GRCh38 aligned CRAM files for WGS (accession EGAS50000000346), and IDAT files for Illumina EPIC methylation array profiling (accession EGAS00001007844) are available from the European-genome-phenome (EGA) archive. As genomic data represent potentially identifying information, these data are under restricted access controlled by the Data Access Committee at the University of Melbourne Centre for Cancer Research. Access to sequence read-level data or germline variant data will require evidence of institutional human research ethics committee approval. Access can be obtained by application through the EGA portal and applications will be processed within four to six weeks. If approved, data access will be granted in perpetuity. Unrestricted access to a data package containing non-identifiable data from tertiary analysis including somatic small and structural variant calls, copy-number analysis, RNA-fusion calls, and gene expression matrices from bulk and single nuclei analysis is available from Figshare (https://doi.org/10.6084/m9.figshare.25792479). RNA-seq, Illumina Infinium HumanMethylation450 arrays, and small-RNA-seq data previously made available by Fishbein et al.[13] was obtained from the National Computational Infrastructure's Genomic Data Commons. Bulk RNA-seq data previously published by Flynn et al.[14] and snRNA data published by Zethoven et al.[22] was obtained from EGA (EGAS00001005861 [https://ega-archive.org/studies/EGAS00001005861]). Small-RNA-seq previously published by Castro-Vega et al. were obtained from ArrayExpress (E-MTAB-2833 [https://www.ebi.ac.uk/arrayexpress/experiments/E-MTAB-2833])[20]. Illumina Infinium HumanMethylation450 and HumanMethylation27 arrays previously made available by Letouzé et al. were obtained from the Gene Expression Omnibus (accession GSE39198, and GSE43293)[9]. The remaining data are available within the Article, Supplementary Information, or Source Data file. Source data are provided with this paper.

## Code availability

All code used in the generation and analysis of data contained within this publication is available from GitHub: https://github.com/UMCCR-RADIO-Lab/a5_sdhb_pcpg (https://doi.org/10.5281/zenodo.14890696)[102].

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

## Acknowledgements

The Clinical Genomics Platform at University of Melbourne Centre for Cancer Research for their support in generating primary data. Leah Meuter for her crucial work in curating clinical data for patients from the NIH. This work was supported by funds from the National Health and Medical Research Project Grant (APP1108032), Pheo Para Alliance (USA), Paradifference Foundation (Sweden) and The SDHB Coalition (USA). RWT was supported by a Victorian Cancer Agency Mid-Career Fellowship (TP828750). A.F. and A.Pa were supported by the Joseph Herman Trust at the University of Melbourne. K.P. was supported by the Intramural Research Program of the NICHD, NIH. P.L.M.D. is a Robert Tucker

Hayes Distinguished Chair in Oncology and was supported by funds the NIH/NIGMS (GM114102), NIH/NCI (CA264248), Neuroendocrine Tumour Research Foundation, VHL Alliance Investigator Award, and Para-difference Foundation. L.F. was supported in part by ACS MRSG-15-063-01. T.D. and R.C.B. were supported by grants from Perpetual (Hillcrest) Foundation. The Novo Nordisk Foundation Center for Stem Cell Medicine, reNEW, is supported by a Novo Nordisk Foundation grant number NNF21CC0073729.

## Author contributions

R.W.T., K.P., and R.C.B. conceived the study. A.F., A.D.P., S.B., E.B., B.B., T.D., F.R., L.M., M.Z. and J.R.N. performed experiments and data analysis. A.S.T. and A.J.G. performed pathology review and analysis. L.S.K., T.E., D.I., L.F., A.J.G., O.H., A.S.T., T.G., R.R.R., H.K.G., A.H.T., I.B., M.S.E., J.N.Y.Y., R.J.H., J.C., T.A., T.P., S.G., P.S., P.D., R.C.B., and K.P. provided material support, recruited patients, or curated clinical data. A.F., R.C.B., K.P., and R.W.T., wrote the manuscript. All authors edited and approved the final manuscript.

## Competing interests

The authors declare no competing interests.

## Additional information

[1]Centre for Cancer Research and Department of Clinical Pathology, University of Melbourne, VIC, Australia. [2]Kolling Institute of Medical Research, Royal North Shore Hospital St Leonards NSW, Melbourne, Australia. [3]Murdoch Children's Research Institute, The Royal Children's Hospital, Melbourne, VIC 3052, Australia. [4]Novo Nordisk Foundation Centre for Stem Cell Medicine, Murdoch Children's Research Institute, Melbourne, VIC 3052, Australia. [5]Australian Regenerative Medicine Institute, Monash University, Victoria, Australia. [6]Peter MacCallum Cancer Centre, Melbourne, Australia. [7]Division of Endocrinology, Diabetes and Metabolism, Department of Internal Medicine, The Ohio State University, Columbus, OH, USA. [8]University of Michigan, Ann Arbor, MI, USA. [9]Department of Medicine, Division of Endocrinology, Metabolism, Diabetes, University of Colorado, Aurora, CO, USA. [10]Sydney Medical School, University of Sydney, Sydney, NSW, Australia. [11]NSW Health Pathology, Department of Anatomical Pathology, Royal North Shore Hospital, St Leonards NSW, Sydney, Australia. [12]Tufts Medical Center, Boston, MA, USA. [13]Eunice Kennedy Shriver National Institute of Child Health and Human Development, Bethesda, MD, USA. [14]Children's Medical Research Institute, Faculty of Medicine and Health, The University of Sydney, Westmead, Australia. [15]Sir Peter MacCallum Department of Oncology, University of Melbourne, Melbourne, VIC, Australia. [16]University of Florida and Malcom Randall VA Medical Center, Gainesville, FL, USA. [17]Division of endocrinology and Research Center, Center hospitalier de l'Université de Montréal, Montreal, Canada. [18]Waikato Clinical Campus, University of Auckland, Hamilton, New Zealand. [19]Cancer Genetics Service, National Cancer Center Singapore, Singapore, Singapore. [20]Lee Kong Chian School of Medicine, Nanyang Technological University Singapore, Singapore, Singapore. [21]St Vincent's Dept of Medicine, University of Melbourne, Melbourne, VIC, Australia. [22]Department of Medical Sciences, Uppsala University, Uppsala, Sweden. [23]Department of Surgical Sciences, Uppsala University, Uppsala, Sweden. [24]Div. Hematology and Medical Oncology, Department of Medicine, Mays Cancer Center, University of Texas Health Science Center at San Antonio (UTHSCSA), San Antonio, TX, USA. [25]These authors contributed equally: Roderick Clifton-Bligh, Karel Pacak, Richard W. Tothill. ✉e-mail: roderick.cliftonbligh@sydney.edu.au; karel@mail.nih.gov; rtothill@unimelb.edu.au

