## [Transparent Peer Review file · Nature Communications]

Multi-omic analysis of SDHB-deficient pheochromocytomas and paragangliomas identifies metastasis and treatment-related molecular profiles

Corresponding Author: Professor Richard Tothill

Version 0:

Reviewer comments:

Reviewer #1

(Remarks to the Author)

Aidan Flynn et al. report the most comprehensive multi-omics characterization of SDHB-mutated paraganglioma to date, with particular emphasis on metastasis and treatment-related molecular events. The key findings include i) clear separation of parasympathetic (non-chromaffin) from sympathetic (chromaffin) cells of origin; ii) confirmation of immortalization-related mechanisms (ATRX loss-of-function and TERT activation) as critical players in metastatic progression, including the first report of an AFF4-TERT fusion; iii) MGMT overexpression and MMR deficiency as resistance mechanisms to alkylating agents. Overall, the study is well-conducted and integrates a significant amount of genomic data. These results, while validating recent observations, could be clinically relevant by defining the molecular changes that accompany the progression of SDHB-mutated tumors. Below are some comments to be addressed:

1) In the introduction section, the authors cite references 10-15 to state that "..., most PCPG genomic studies to date have used whole exome sequencing, limiting detection of non-coding variants, structural alterations, telomeric features, and DNA mutation patterns. Furthermore, these studies have predominantly analyzed primary and non-metastatic tumors, thereby limiting the discovery of metastasis biomarkers." I am afraid this is not true at all for studies #11 and #15, which depict a genomic characterization of metastatic paraganglioma (#11) and a thorough assessment of immortalization-related mechanisms in metastatic paraganglioma (#15), respectively.

2) At the end of the introduction section, the authors mention they confirmed telomeric dysfunction and other features in metastatic cases. I suggest changing the term "telomeric dysfunction" to "telomere maintenance mechanisms" because the former implies ongoing chromosome instability from telomeric origin, such as telomere crisis, and this study did not address the origin nor the consequences of telomere dysfunction.

3) Although informative, the authors should acknowledge the limitations of using genomic data to estimate telomere content (which includes interstitial telomere sequences) and TERRA expression (which is fairly low, below 8 reads).

4) Related to the previous comment, a critical aspect of this paper is the assessment of the alternative lengthening of telomere (ALT) mechanism. According to a previous pan-cancer genomic study on immortalization mechanisms, ALT features can include aberrant telomeric variant repeat (TVR) usage, intrachromosomal insertion of telomeric DNA, and expression of telomeric repeat-containing RNA (TERRA). Nevertheless, ATRX mutations, long telomeres, high TERRA expression, or the presence of C-circles are not specific markers to assign the ALT status. It is important to mention that telomere heterogeneity, a definitive feature of ALT, was not evaluated. For instance, the authors found C-circles in one TERT-altered primary tumor, whereas long telomeres in the absence of C-circles were detected in four ATRX/TERT wild-type paragangliomas. Proper ALT annotation is very important for conclusions, acknowledging that i) ALT could be present independently of ATRX mutations, and ii) some metastatic paragangliomas seem to progress in the absence of a telomere maintenance mechanism.

5) Although the authors report transcriptional differences between SDHB-mutated metastatic and SDHB-mutated non-metastatic paragangliomas (highlighting previous findings such as high CDK1), a more integrative analysis of genomic data would be appreciated in this context. For instance, what is the contribution of methylation vs. genome instability to those

expression changes? What about miRNA expression changes? Note also that the overexpression of PRC2 subunits EZH2 and SUZ12 was reported to be significant in SDHB-mutated tumors, but the target genes were more related to TET-inhibition (Morin A et al. Cell Reports 2020). Similarly, snRNA/ATAC-seq data could be more exploited. For instance, what are the most significant transcription factor binding motifs in SDHB-mutated metastatic paragangliomas? I think such integrative analyses would be more relevant for the purpose of the study than comparing transcriptional changes between TERT and ATRX-altered paragangliomas because, once again, without a clear assessment of the ALT phenotype, this is difficult to interpret.

6) In the results section "TERT/ATRX alterations are associated with metastatic progression and late somatic events," the authors should acknowledge that the association of immortalization-related mechanisms with tumor size, which was restricted to telomerase-related tumors, is consistent with a previous study (Ref #15).

7) The association of microsatellite instability with metastatic paraganglioma was highlighted in a previous study (Ref #11). Please discuss this more in the context of alkylating chemotherapy. For instance, while microsatellite instability in MMR-deficient gliomas was not detected by bulk analyses, single-cell whole-genome sequencing of post-treatment hypermutated glioma cells demonstrated microsatellite mutations (Ref #46).

Minor comments:

1. Legend Figure 2: Words DNA and data are in bold

2. Lines 129-130: Rephrase "By DNA methylation profiling, SDHB-mutant PCPG independently of other PCPG genotypes with the exception of one tumour (E229-P1)" to "By DNA methylation profiling, SDHB-mutant PCPG was distinguished from other PCPG genotypes, with the exception of one tumor (E229-P1)".

3. Lines 178-179: "A minority of tumors had chromothripsis (n=6) or genome doubling (n=9), as previously reported in PCPG [13,14]. The authors are missing Ref #20, which indeed corresponds to the first report of chromothripsis in paraganglioma.

Reviewer #2

(Remarks to the Author)

In this work, Flynn et al. generated a large multi-omic dataset comprising genomic, transcriptomic, epigenomic and functional data on 94 tumors from 79 patients with SDHB-mutant pheochromocytomas and paragangliomas (PCPG). They used this resource to characterize molecular alterations in this aggressive PCPG subgroup, and to identify those associated with metastatic disease.

This is overall a very interesting study. The dataset, integrating 7 molecular methods on a large cohort with both primary and metastatic SDHB-mutated PCPG, is extremely valuable. The analysis is sound and the paper is well-written. The results, nicely supported by the data, reinforce previously described associations in a larger cohort, and reveal important new findings notably related to the rare head and neck subgroup and potential mechanisms of resistance to DNA alkylating therapies.

I have a few questions and suggestions:

1) In Fig. 1D, the authors display the diversity of genomic alterations in SDHB. Is there an association between the type of SDHB alteration (missense mutation vs. truncative, location etc.) and the risk of metastasis?

2) The authors report chromothripsis and genome doubling in a few cases. Are these features associated with increased metastasis risk?

3) Single-nucleus data could be valuable to dissect intra-tumor heterogeneity in PCPG. It seems to me that these data are underused in the study. Could the authors comment on the added value of snRNA-seq and snATAC-seq? Did they identify heterogeneous cell types with specific transcriptomic / chromatin accessibility landscapes?

4) The authors analyzed multiple samples from ~10 patients. Some examples of molecular evolution are described, e.g. the late development of TERTp mutations, or the two patients with samples pre / post-CVD. However, I believe such data or so valuable that it would be worth reporting the molecular evolution of all cases with multiple samples. In particular, for patients with metastatic disease, were there other driver mutations / CNAs, or transcriptomic / epigenetic events acquired or expanded at metastasis (in addition to TERTp mutations)?

5) Hypermethylator phenotype is an important feature of SDHB-mutated PCPG. How does this phenotype evolve in metastatic vs. primary tumors? Is it reinforced? Do new loci become hypermethylated? Is the intensity of the hypermethylator phenotype associated with a higher risk of metastasis?

6) In Fig. 8 (and other examples of molecular evolution), I would suggest showing oncogenetic trees instead of shared / private variants to give a more intuitive view of early / late events.

Reviewer #3

(Remarks to the Author)

The manuscript by Flynn et al., have conducted extensive multi-omics analysis on SDHB-deficient PCPGs, which show relatively high metastasis rate. The authors have characterized the PCPGs with WGS, WTS, small RNA-seq, methylation-seq, with some snATAC/RNA-seq analysis. This allowed them to disentangle detailed genotypic and molecular subtypes, especially distinguishing non-chromaffin HN-PGs from sympathetic PCPGs. The authors especially focused on the relationship between TLM, ALT and metastasis, showing that TERT and ATRX mutations are mutually exclusive, with ATRX mutation showing ALT phenotype coupled with the presence of c-circle. These mutations are correlated with better proliferation score and metastatic cancers. Finally, they show the genomic, transcriptomics changes upon chemotherapy, revealing some potential targets conferring drug resistance.

This study features extraordinary data collection on PCPGs at multiple levels, with thorough bioinformatic analysis. The data is well organized and presented. While I'm generally positive for the publication of the manuscript, there are some options for the improvement:

(1) Validation experiments:

Based on transcriptome analysis, the authors suggest potential mechanism for cell cycle regulation in ATRX mutants, and drug resistance in chemo-treated samples. If the authors could provide additional cell line-based K/D or K/O experiments on those suggested targets, it would improve the impact and novelty of paper

(2) More description on TLM without TERT/ATRX mutations:

The authors have noted E158 and E159 patients which showed metastasis without TERT/ATRX mutations. Could the authors provide more detailed description on these patients which could provide hint on TERT/ATRX independent TLM and metastasis?

(3) From the transcriptome and epigenome (snATAC and methylation) analysis, could the authors find any evidence on ATRX mutations promoting survival/metastasis independent of ALT?

(4) The authors have noted the increased TMB for the TERT and ATRX mutants. From the genome analysis, could they suggest whether TMB increase precedes the TERT/ATRX mutations or it is followed after the mutation? Perhaps, we could exclude the ATRX mutant cases, and then draw scatter plot comparing TMB vs TERT gene expression?

(5) TERRA analysis: Could the authors describe more about how they handled the strandness of TERRA transcript for the quantification?

Minor suggestions on slightly confusing expressions:

- Normalized to total read count only three TVRs (TTAGGG, GTTGGG, GTAGGG) remained significant after false-discovery rate correction (Benjamini & Hochberg, $p < 0.1$).

- Meanwhile, few genes were uniquely expressed in TERT-altered tumours but included IRX3, SDK1 and TRIP13 (Figure 7E).

- Only one intrachromosomal telomeric insertion was detected in an ATRX altered case (data not shown), while TERRA expression was elevated in ATRX altered tumours, although was not statistically significant (Figure 5D).

Version 1:

Reviewer comments:

Reviewer #1

(Remarks to the Author)

None

(Remarks on code availability)

Reviewer #2

(Remarks to the Author)

The authors have satisfactorily answered to all my questions and comments. I have no other remark and I recommend this paper for acceptance.

(Remarks on code availability)

The Github folder is well organized and documented.

It seems to contain the relevant codes to reproduce the analysis, but I honestly don't have the time to go through all the scripts.

REVIEWER COMMENTS

Reviewer #1, expertise in pheochromocytomas and paragangliomas omics (Remarks to the Author):

Aidan Flynn et al. report the most comprehensive multi-omics characterization of SDHB-mutated paraganglioma to date, with particular emphasis on metastasis and treatment-related molecular events. The key findings include i) clear separation of parasympathetic (non-chromaffin) from sympathetic (chromaffin) cells of origin; ii) confirmation of immortalization-related mechanisms (ATRX loss-of-function and TERT activation) as critical players in metastatic progression, including the first report of an AFF4-TERT fusion; iii) MGMT overexpression and MMR deficiency as resistance mechanisms to alkylating agents. Overall, the study is well-conducted and integrates a significant amount of genomic data. These results, while validating recent observations, could be clinically relevant by defining the molecular changes that accompany the progression of SDHB-mutated tumors. Below are some comments to be addressed:

1) In the introduction section, the authors cite references 10-15 to state that "..., most PCPG genomic studies to date have used whole exome sequencing, limiting detection of non-coding variants, structural alterations, telomeric features, and DNA mutation patterns. Furthermore, these studies have predominantly analyzed primary and non-metastatic tumors, thereby limiting the discovery of metastasis biomarkers." I am afraid this is not true at all for studies #11 and #15, which depict a genomic characterization of metastatic paraganglioma (#11) and a thorough assessment of immortalization-related mechanisms in metastatic paraganglioma (#15), respectively.

Reply: We have removed the final sentence from the paragraph "many of these genomic sequencing studies have predominantly analysed primary and non-metastatic tumours thereby limiting discovery of metastasis biomarkers".

2) At the end of the introduction section, the authors mention they confirmed telomeric dysfunction and other features in metastatic cases. I suggest changing the term "telomeric dysfunction" to "telomere maintenance mechanisms" because the former implies ongoing chromosome instability from telomeric origin, such as telomere crisis, and this study did not address the origin nor the consequences of telomere dysfunction.

Reply: We have changed the text to state confirmation of mutations in *ATRX* and *TERT*. This accounts for the potential non-telomere maintenance mechanisms associated with these gene mutations (line 89).

3) Although informative, the authors should acknowledge the limitations of using genomic data to estimate telomere content (which includes interstitial telomere sequences) and TERRA expression (which is fairly low, below 8 reads).

Reply: While we understand that indirect measurement of telomere content by tag counting may not be as precise as other experimental methods, the authors of TelomereHunter benchmarked their tool against a qPCR based method and demonstrated a Spearman correlation of 0.91 (PMID: 31138115). Given the dynamic range of our data and the strong agreement with laboratory-based methods, we feel that tag counting provides sufficient

accuracy to segregate our samples broadly into those with long and short telomeres as we have done.

Regarding interstitial telomere sequences, we searched for intrachromosomal telomeric DNA insertions using breakpoint detection methods but found limited evidence for this occurring in PCPG tumours unlike in other cancer types (PMID: 25723166). Interstitial telomere sequences occurring in the centromeric region that consist of entirely telomeric repeats are difficult to differentiate from true telomeric reads, as such we have added a remark highlighting this limitation (line 240-242).

TERRA detection was performed by applying TelomereHunter to RNA-seq data. We generated between 100 and 200 million reads per case and we saw a median of 1524 TERRA reads per sample. Therefore, we are unsure what the reviewer is referring to regarding TERRA counts of eight reads.

4) Related to the previous comment, a critical aspect of this paper is the assessment of the alternative lengthening of telomere (ALT) mechanism. According to a previous pan-cancer genomic study on immortalization mechanisms, ALT features can include aberrant telomeric variant repeat (TVR) usage, intrachromosomal insertion of telomeric DNA, and expression of telomeric repeat-containing RNA (TERRA). Nevertheless, *ATRX* mutations, long telomeres, high TERRA expression, or the presence of C-circles are not specific markers to assign the ALT status. It is important to mention that telomere heterogeneity, a definitive feature of ALT, was not evaluated. For instance, the authors found C-circles in one *TERT*-altered primary tumor, whereas long telomeres in the absence of C-circles were detected in four *ATRX/TERT* wild-type paragangliomas. Proper ALT annotation is very important for conclusions, acknowledging that i) ALT could be present independently of *ATRX* mutations, and ii) some metastatic paragangliomas seem to progress in the absence of a telomere maintenance mechanism.

Reply: We have performed a comprehensive genome analysis to detect known features of ALT utilising WGS (e.g. total telomeric repeat count and TVR usage), WTS (TERRA expression) as well as an independent assay to detect C-circles, which is a well validated hallmark of ALT (PMID: 19935656). A detailed study of ALT features by one of the co-authors, currently being prepared for publication, indicates that the C-circle assay is currently the most robust independent indicator of ALT status. The ALT-FISH and ALT-associated PML body (APB) assays, like the C-circle assay, detect extrachromosomal telomeric DNA, and rarely add any additional information. Heterogeneity of telomere length is highly correlated with a positive C-circle assay, but is usually assessed by Southern blotting of terminal restriction fragments and requires the availability of microgram quantities of high molecular weight genomic DNA which often makes it an unsuitable method for detection of ALT in human tumour cohorts.

All *ATRX* mutant tumours in our cohort had hallmarks of ALT including an increase in telomeric repeats (over blood DNA) and C-circles; however, some genomic features such as intrachromosomal telomere insertions and TVR usage were not nearly as prominent as reported in some other cancer types (PMID: 32024817). This suggests that some genomic ALT features may be dependent on cancer type, as previously reported and cannot be relied upon to classify ALT+ tumours (PMID: 33692341)

5) Although the authors report transcriptional differences between *SDHB*-mutated metastatic

and SDHB-mutated non-metastatic paragangliomas (highlighting previous findings such as high CDK1), a more integrative analysis of genomic data would be appreciated in this context. For instance, what is the contribution of methylation vs. genome instability to those expression changes?

Reply: We did not see any association between copy-number and differential expression among top genes DE between metastatic and non-metastatic tumours using Pearson correlation < 0.07 (Figure R1A). In Figure 7C and E we have shown many of the under-expressed genes were also within in DMR regions, however, the relationship between methylation and expression is complex and highly variable between CpG probes in proximity to a gene and variable associations are found with respect to 5mC and expression for different genes (Figure R1B).

Figure R1
For each of the top genes differentially expressed between metastatic tumours (primary/metastases) and non-metastatic primary samples: **(A)** Pearson correlation (R^2) between gene copy number and gene expression (\log_2 CPM). **(B)** Pearson correlation (R^2) between probe methylation (M-value) for each probe (dots) associated with a given gene and gene expression (\log_2 CPM). The median correlation across all probes for a given gene is indicated by a grey bar.

Reviewer 1 Point 5 continued. What about miRNA expression changes?

Reply: We have performed differential expression analysis for small RNA contrasting metastatic and non-metastatic tumours, *ATRX*-mutant vs non-metastatic, *TERT*-mutant vs non-metastatic and *TERT*-mutant vs *ATRX*-mutant tumours. A new supplementary figure (Supplementary Figure 18) has now been added describing these findings and text added accordingly to the results section (lines 300-308). We found overexpression of three miRNAs (hsa-miR-96-5p, hsa-miR-183-5p, hsa-miR-182-5p) in *TERT/ATRX* tumours,

consistent with previous reports that these miRNAs are a marker of metastatic behaviour in PCPG (PMID: 31410193) and we have now also included this citation.

Reviewer 1 Point 5 continued. Note also that the overexpression of PRC2 subunits EZH2 and SUZ12 was reported to be significant in SDHB-mutated tumors, but the target genes were more related to TET-inhibition (Morin A et al. Cell Reports 2020).

Reply: Thank you for highlighting this paper. As Morin et al also showed a difference in EZH2 expression between metastatic and benign tumours we have included a citation for this in our manuscript.

Reviewer 1 Point 5 continued. Similarly, snRNA/ATAC-seq data could be more exploited. For instance, what are the most significant transcription factor binding motifs in SDHB-mutated metastatic paragangliomas?

Reply: We have now performed additional analysis looking at enrichment of TF binding sites. We have now included a new Supplementary Figure 19 and have added appropriate text to the Results section in the manuscript (lines 310-319)

Reviewer 1 Point 5 continued. I think such integrative analyses would be more relevant for the purpose of the study than comparing transcriptional changes between TERT and ATRX-altered paragangliomas because, once again, without a clear assessment of the ALT phenotype, this is difficult to interpret.

Reply: Given a) the link between *ATRX* mutations and metastatic disease rather than ALT features and metastatic disease and b) *ATRX* can have functions independent of TMM we believe it was more important to focus on the biology and molecular correlates of *ATRX* and TERT mutations (e.g. gene-expression) rather than to focus on the ALT phenotype.

6) In the results section "TERT/*ATRX* alterations are associated with metastatic progression and late somatic events," the authors should acknowledge that the association of immortalization-related mechanisms with tumor size, which was restricted to telomerase-related tumors, is consistent with a previous study (Ref #15).

Reply: We have now cited this reference in the Results section discussing association between primary tumour size and *TERT/ATRX* mutations.

7) The association of microsatellite instability with metastatic paraganglioma was highlighted in a previous study (Ref #11). Please discuss this more in the context of alkylating chemotherapy. For instance, while microsatellite instability in MMR-deficient gliomas was not detected by bulk analyses, single-cell whole-genome sequencing of post-treatment hypermutated glioma cells demonstrated microsatellite mutations (Ref #46).

Reply: We did not find evidence of MSI in metastatic paragangliomas without exposure to DNA alkylating chemotherapy. The MSI score was low and equivalent between metastatic and non-metastatic tumours aside from the one case that had been exposed to dacarbazine and importantly had also acquired a deleterious *MLH1* mutation consistent with resistance to DNA alkylating chemotherapy (see Supplementary Figure 21).

We agree that bulk tissue analysis may be limited in detecting MSI in minor subclones and that the detection of MSI or SBS11 would require sufficient time for outgrowth of the resistant

clone. It is possible that in some tumours that were exposed to DNA alkylating chemotherapy may have had minor resistant subclones yet we could not detect signature 11 or MSI because of insufficient depth or tumour sampling.

Minor comments:

1. Legend Figure 2: Words DNA and data are in bold

Reply: The bolding has been removed from these words

2. Lines 129-130: Rephrase "By DNA methylation profiling, SDHB-mutant PCPG independently of other PCPG genotypes with the exception of one tumour (E229-P1)" to "By DNA methylation profiling, SDHB-mutant PCPG was distinguished from other PCPG genotypes, with the exception of one tumor (E229-P1)".

Reply: We have integrated the suggested correction.

3. Lines 178-179: "A minority of tumors had chromothripsis (n=6) or genome doubling (n=9), as previously reported in PCPG [13,14]. The authors are missing Ref #20, which indeed corresponds to the first report of chromothripsis in paraganglioma.

Reply: We have now included ref#20 as a citation for chromothripsis.

Reviewer #2, expertise in pheochromocytomas and paragangliomas omics (Remarks to the Author):

In this work, Flynn et al. generated a large multi-omic dataset comprising genomic, transcriptomic, epigenomic and functional data on 94 tumors from 79 patients with SDHB-mutant pheochromocytomas and paragangliomas (PCPG). They used this resource to characterize molecular alterations in this aggressive PCPG subgroup, and to identify those associated with metastatic disease.

This is overall a very interesting study. The dataset, integrating 7 molecular methods on a large cohort with both primary and metastatic SDHB-mutated PCPG, is extremely valuable. The analysis is sound and the paper is well-written. The results, nicely supported by the data, reinforce previously described associations in a larger cohort, and reveal important new findings notably related to the rare head and neck subgroup and potential mechanisms of resistance to DNA alkylating therapies.

I have a few questions and suggestions:

1) In Fig. 1D, the authors display the diversity of genomic alterations in SDHB. Is there an association between the type of SDHB alteration (missense mutation vs. truncative, location etc.) and the risk of metastasis?

Reply: We had previously looked at mutation position in the gene and clinical behaviour and found no significant association (Fisher's exact test comparing mutations occurring before and after amino acid 140 separating the iron cluster protein domains, p-value=1). We have now compared clinical behaviour between missense and nonsense mutation types and found no significant difference (Fisher's exact test, p-value = 0.423). There is insufficient data to compare between specific mutations or large deletions.

2) The authors report chromotripsis and genome doubling in a few cases. Are these features associated with increased metastasis risk?

Reply: We found evidence of chromothripsis in both metastatic and non-metastatic tumours; however, there are too few cases with chromothripsis to conclude any enrichment within either the metastatic or non-metastatic group.

3) Single-nucleus data could be valuable to dissect intra-tumor heterogeneity in PCPG. It seems to me that these data are underused in the study. Could the authors comment on the added value of snRNA-seq and snATAC-seq? Did they identify heterogenous cell types with specific transcriptomic / chromatin accessibility landscapes?

Reply: We have now included a supplementary figure exploring subclonal heterogeneity of copy number events using both WGS and single-nuclei RNA-seq (Supplementary Figure 14). This is now called out in the results (lines 182-185). Although clonal heterogeneity is apparent the biological and clinical significance of intratumoural heterogeneity is difficult to interpret in most cases, However, we have highlighted an example in the current study where subclonal expansion of *TERT* mutation does occur and gives rise to metastases, which is perhaps the most biologically and clinically relevant observation from our data.

4) The authors analyzed multiple samples from ~10 patients. Some examples of molecular evolution are described, e.g. the late development of TERTp mutations, or the two patients with samples pre / post-CVD. However, I believe such data is so valuable that it would be worth reporting the molecular evolution of all cases with multiple samples. In particular, for patients with metastatic disease, were there other driver mutations / CNAs, or transcriptomic / epigenetic events acquired or expanded at metastasis (in addition to TERTp mutations)?

Reply: We have constructed evolutionary trees for all primary/metastasis pairs and included them in supplementary figure 2 with a callout in the text (line 102). We have also annotated the list of somatic coding variants (supplementary data 5) with a private/shared status according to whether they were observed in paired samples (where available). There are many mutations which are either shared between the primary and metastatic tumour or private to the metastasis, however there is very little recurrence beyond *TERT*. As such, interpreting the importance of these single-observation variants is difficult.

Please note that as part of the new clonal evolution analysis we discovered a number of mutations with low but similar variant allele fractions across paired primary and metastatic tumours. As this is biologically unlikely, we suspected these were potentially false-positive variant calls previously missed. Indeed, we found many of these variants were recurrent in unrelated cases but had been rejected as false positives in these samples. We therefore added another layer of variant filtration (see updated methods) to remove these variants based on how often they were observed in other tumours and rejected by the variant callers.

This has resulted in small modifications to several of the figures but does not change the interpretation of any of the data presented.

5) Hypermethylator phenotype is an important feature of SDHB-mutated PCPG. How does this phenotype evolve in metastatic vs. primary tumors? Is it reinforced? Do new loci become hypermethylated? Is the intensity of the hypermethylator phenotype associated with a higher risk of metastasis?

Reply: Examining the proportion of probes which are methylated (beta-value > 0.7) does not show any significant difference between non-metastatic primaries and metastasis (Figure R2). In cases where we had paired samples, the number of probes which increased or decreased in methylation between the primary and metastasis seems to be highly variable, with some pairs showing more probes becoming demethylated than methylated, and others showing the opposite (Figure R3). It is therefore difficult with the data we have to draw any overarching conclusions about the relative methylation state between primary and metastatic tumours.

Figure R2
The percentage of EPIC methylation array probes with a beta value above 0.7 (y-axis) for each sample (dots). The clinical behaviour of the sample is indicated on the x-axis and by color. No groups were significantly different to any other by Student's t-test.

Figure R3 Relative beta levels of identical probes between paired samples

Methylation levels of matched probes were compared between paired and unrelated (far right) samples. In each comparison, probes in the second sample with a beta value more than 0.25 above or below the the beta value in the first sample were considered increased or decreased, respectively. Probes with a beta value within 0.25 were considered unchanged.

6) In Fig. 8 (and other examples of molecular evolution), I would suggest showing oncogenetic trees instead of shared / private variants to give a more intuitive view of early / late events.

Reply: We have converted the shared/private variant diagrams into tree style plots to provide greater clarity in Figure 8. We have additionally provided evolutionary analysis for all primary/metastasis pairs in Supplementary Figure 2, as previously requested.

Reviewer #3, expertise in single cell multi-omics (Remarks to the Author):

The manuscript by Flynn et al., have conducted extensive multi-omics analysis on SDHB-deficient PCPGs, which show relatively high metastasis rate. The authors have characterized the PCPGs with WGS, WTS, small RNA-seq, methylation-seq, with some snATAC/RNA-seq analysis. This allowed them to disentangle detailed genotypic and molecular subtypes, especially distinguishing non-chromaffin HN-PGs from sympathetic PCPGs. The authors especially focused on the relationship between TLM, ALT and metastasis, showing that *TERT* and *ATRX* mutations are mutually exclusive, with *ATRX* mutation showing ALT phenotype coupled with the presence of c-circle. These mutations are correlated with better proliferation score and metastatic cancers. Finally, they show the genomic, transcriptomics changes upon chemotherapy, revealing some potential targets conferring drug resistance.

This study features extraordinary data collection on PCPGs at multiple levels, with thorough bioinformatic analysis. The data is well organized and presented. While I'm generally positive for the publication of the manuscript, there are some options for the improvement:

(1) Validation experiments:

Based on transcriptome analysis, the authors suggest potential mechanism for cell cycle

regulation in *ATRX* mutants, and drug resistance in chemo-treated samples. If the authors could provide additional cell line-based K/D or K/O experiments on those suggested targets, it would improve the impact and novelty of paper

Reply: While we agree that functional studies will be important, we believe these experiments really lie outside the scope of the current study given the focus is on a genomic landscape analysis. Furthermore, there is a paucity of good cell line models of SDHB-deficient PCPG and it is not possible to generate such data within a reasonable timeframe.

(2) More description on TLM without TERT/*ATRX* mutations:

The authors have noted E158 and E159 patients which showed metastasis without TERT/*ATRX* mutations. Could the authors provide more detailed description on these patients which could provide hint on TERT/*ATRX* independent TLM and metastasis?

Reply: We have explored these cases quite extensively and were unable to find any events that were likely to be alternative drivers of metastasis. There were no common coding mutations or gene expression outliers between the two cases making it quite difficult to draw conclusions from a single observation.

(3) From the transcriptome and epigenome (snATAC and methylation) analysis, could the authors find any evidence on *ATRX* mutations promoting survival/metastasis independent of ALT?

Reply: Using our transcriptome data we showed *ATRX*-mutant tumours showed upregulation of several genes linked to cell division (*TLX1*, *RPRM*, *DRG2*) or metastasis (*OTX1*) as well as several other genes listed in the manuscript. Additionally, *ATRX*-mutant tumours showed increased expression of non-coding RNAs as well as miRNAs targeting a known inhibitor of metastasis (*RECK*).

(4) The authors have noted the increased TMB for the TERT and *ATRX* mutants. From the genome analysis, could they suggest whether TMB increase precedes the TERT/*ATRX* mutations or it is followed after the mutation? Perhaps, we could exclude the *ATRX* mutant cases, and then draw scatter plot comparing TMB vs TERT gene expression?

Reply: It is quite difficult to tease apart these two scenarios with the data we have because in most cases *TERT* mutations are clonal in both the primary and metastasis. The two cases which provide some insights are a wild-type primary with a paired metastasis harbouring a *TERT* mutation (E146), and a primary with a sub-clonal *TERT* mutation which becomes dominant in the metastasis (E143). In the first case the tumour mutation burden of the wildtype primary is lower than the metastasis (0.38 vs 0.82). In the second case, clonal evolution analysis reveals 258 ancestral mutations shared across all clones, 640 private mutations associated with non-metastatic clones in the primary, and 836 mutations associated with the metastatic (*TERT*-mutant) clones (Supplementary Figure 2). As the *TERT* mutation was only present in 15% of reads (purity adjusted) the non-metastatic clones likely comprise around 70% of the tumour but contribute less than 50% of the mutations. While together these data may suggest that *TERT*-mutations lead to a higher rate of mutation, one should consider that these cells may have undergone more cell divisions and one may expect more mutations simply through DNA replication errors rather than an additional TER-related mutational process. Besides the number of additional mutations is not large and the tumour mutation burden is still very low. The assumption that increased

mutations is due to additional cell divisions is partially supported by the observation that *TERT*-mutant tumours are associated with larger primary tumours (Figure R4). Unfortunately, there is insufficient data to make a firmer comment as to whether the higher TMB in *TERT*-mutant cases represents cause or an effect. We have included a plot of TMB versus *TERT*-expression for the reviewers' interest (Figure R5)

Figure R4
Tumour mutation burden (y-axis, mutations per megabase) versus the largest dimension of the of the largest primary tumour observed in the patient (x-axis, centimetres). Point colour indicates the presence or absence of a *TERT* or *ATRX* mutation. Samples from the same patient are joined with a dotted line.

Figure R5
TERT expression (x-axis, log₂ CPM) versus tumour mutation burden (y-axis, mutations per megabase). Point colour indicates the clinical behaviour of the tumour specimen and shape indicates the presence of a *TERT* or *ATRX* mutation. Samples from the same patient are joined with a dotted line.

(5) TERRA analysis: Could the authors describe more about how they handled the strandness of TERRA transcript for the quantification?

Reply: While we did utilise a strand-specific RNA-sequencing library preparation, we did not use a strand-aware bioinformatic approach when performing tag counting. To our

knowledge, there is no reason why TERRA being transcribed from the C-rich strand should bias the tag counting.

Minor suggestions on slightly confusing expressions:

- Normalized to total read count only three TVRs (TTAGGG, GTTGGG, GTAGGG) remained significant after false-discovery rate correction (Benjamini & Hochberg, $p < 0.1$).

Reply: We have modified this sentence to read

“When normalized to total read count (rather than telomeric read count) only three TVRs (TTAGGG, GTTGGG, GTAGGG) remained significant after false-discovery rate correction (Benjamini & Hochberg, $p < 0.1$)” (Lines 234-235)

- Meanwhile, few genes were uniquely expressed in TERT-altered tumours but included IRX3, SDK1 and TRIP13 (Figure 7E).

Reply: We have altered this sentence to read:

“Meanwhile, few genes were uniquely expressed in *TERT*-altered tumours; amongst these were IRX3, SDK1 and TRIP13 (Figure 7E)” (Line 298)

- Only one intrachromosomal telomeric insertion was detected in an ATRX altered case (data not shown), while TERRA expression was elevated in ATRX altered tumours, although was not statistically significant (Figure 5D).

Reply: We have broken up these sentences to improve clarity. The text now reads:

“Only one intrachromosomal telomeric insertion was detected across the cohort, occurring in *ATRX* altered case E198-M1 (data not shown). TERRA expression was elevated in *ATRX* altered tumours, however, the difference did not reach statistical significance (Two-sided Student’s t-test $p > 0.05$)(Figure 5D).” (Lines 238-240)